# Position: Regulating Algorithms Is Not Enough. A Study on Content Discovery in Online Platforms

Rebecca Salganik [1]   Guillaume Salha-Galvan [2]   Adelaida Afilipoaie [3]   Gustavo Ferreira [4]   Valdy Wiratama [5]
Anson Kahng [1]   Jian Kang [6]   Heritiana Ranaivoson [3]

## Abstract

Recent AI regulation has largely focused on algorithmic components such as recommender models, ranking systems, and profiling mechanisms. At the same time, cultural and digital policy agendas increasingly frame discovery as a key objective, aiming to promote exposure diversity and cultural representation. We argue that these outcomes cannot be effectively governed through algorithm-centric approaches alone. Discovery does not arise from individual algorithms in isolation, but from interactions among models, interfaces, user behavior, economic incentives, and cultural norms. We introduce the Cultural Expressions Discovery Circuit (CEDC), an interdisciplinary framework that models discovery as an emergent socio-technical process. Through this lens, we illustrate how certain regulatory approaches struggle to align with broader cultural objectives. Finally, we highlight how socio-technical analysis can help inform both technical research and the governance of cultural expressions in online platforms.

## 1. Introduction

Regulatory efforts targeting artificial intelligence (AI) and machine learning (ML) systems have intensified across multiple jurisdictions in recent years, motivated by concerns around transparency, fairness, accountability, and societal impact (European Parliament, 2024; Ministère de la Culture (France), 2025). Within these initiatives, algorithmic systems–particularly those used for ranking, personalization, and recommendation–have become central objects of governance, with legal obligations shaped by longstanding academic debates on cultural mediation and platform power (Born et al., 2021; Prey, 2020a; Ranaivoson, 2010).

The influence of algorithmic recommendation systems on cultural visibility, attention, and consumption has been widely documented across media studies, cultural sociology, and human–computer interaction (Hesmondhalgh, 2019; Gillespie, 2016; Ranaivoson & Domazetovikj, 2023; Garcia-Gathright et al., 2018; McKelvey & Hunt, 2019; Loh, 2022). In response, legislators have increasingly articulated *discovery* and *discoverability* as explicit policy objectives (European Parliament, 2024; Ministère de la Culture (France), 2025; European Commission, 2024). While these concepts lack universally agreed-upon definitions, they are commonly treated as interdependent: discovery refers to the process through which users encounter and engage with novel content, while discoverability concerns the likelihood that such content is surfaced and made available in the first place. In cultural policy contexts discoverability has become a key mechanism for promoting diversity, exposure breadth, and access to niche cultural expressions (Ranaivoson, 2010; Ranaivoson & Domazetovikj, 2023; European Parliament, 2024).

Crucially, much of the regulatory discourse in this domain remains framed in algorithm-centric terms, focusing on discrete technical components such as models, profiling mechanisms, explainability requirements, or documentation of automated decision-making (Ulbricht & Yeung, 2024a; Hesmondhalgh et al., 2023). This framing assumes that imposing obligations at the level of individual algorithms is sufficient for manifesting broader improvements in diversity, representation, or fairness. In doing so, it implicitly treats algorithms as self-contained systems whose impacts can be understood, audited, and corrected in isolation (Loring, 2025; Ulbricht & Yeung, 2024a) and this, in turn, informs the way that the broader AI community approaches the design of such systems. We argue that this assumption is increasingly misaligned with the socio-technical reality of contemporary ML systems, particularly in the space of cultural expressions and creative content (Dolata et al., 2022). Importantly, these

[1] University of Rochester, New York, USA [2] Shanghai Jiao Tong University, Shanghai, China [3] Vrije Universiteit Brussel, Brussels, Belgium [4] University of Toronto, Toronto, Canada [5] Universität Salzburg, Salzburg, Austria [6] Mohamed bin Zayed University of Artificial Intelligence, Abu Dhabi, United Arab Emirates. Correspondence to: Rebecca Salganik <rsalgani@ur.rochester.edu>, Heritiana Ranaivoson <Heritiana.Renaud.Ranaivoson@vub.be>.

*Proceedings of the 43rd International Conference on Machine Learning*, Seoul, South Korea. PMLR 306, 2026. Copyright 2026 by the author(s).

ecosystems are also shaped by economic and competitive dynamics. Platform operators optimize not only for technical performance metrics, but for business objectives tied to engagement, retention, advertising, subscription growth, and market differentiation (Raff et al., 2020; Drott, 2018). At the same time, creators and cultural producers adapt strategically to platform incentives, while competition between platforms influences catalog design, visibility mechanisms, and discovery strategies (Born et al., 2021; Hesmondhalgh et al., 2023; Prey, 2020b). These forces affect both the supply of cultural content and the behavioral data on which algorithmic systems rely. Meanwhile, even advanced algorithmic techniques act only on existing content and are unable to directly influence these exogenous factors. For example, re-ranking content to promote independent artists can increase short-term exposure, but does not address structural constraints such as limited production resources, lack of audience information, or coordination barriers among creators (Born et al., 2021; Hesmondhalgh et al., 2023; Prey, 2020b). Consequently, discovery outcomes emerge not only from technical design choices, but from the co-evolution of algorithms, markets, and the cultural production systems surrounding them.

Our central claim is that **a holistic understanding of how components interact within an algorithmic ecosystem is essential for governing and designing AI systems that can meaningfully support discovery of cultural expressions.** In practice, outcomes of regulatory interest emerge not from algorithms alone, but from the interaction of models, interfaces, data practices, platform incentives, user behavior, and downstream feedback loops. We show that purely, algorithm-centric governance risks creating structural blind spots, enabling formal compliance without meaningful change to cultural outcomes (Ananny & Crawford, 2018; Loring, 2025; Born et al., 2021). Ultimately, we highlight that effective regulation over a broad spectrum of exogenous effects from ML-driven platforms cannot operate solely at the level of individual algorithms, especially when policy objectives concern broad outcomes such as discovery, diversity, or cultural exposure.

To achieve this, we introduce the *Cultural Expressions Discovery Circuit* (CEDC), a conceptual framework that models content discovery as an emergent property of a socio-technical ecosystem composed of many diverse algorithmic and cultural components. We show that discovery of cultural expressions arises from interactions between components, where each individual outcome propagates and reshapes behavior elsewhere. Ultimately, the CEDC framework clearly maps these interactions between components, providing a map for academics and policy makers to better diagnose research and regulatory targets for key issues related to diversity, exposure, and cultural representation.

The remainder is paper is structured around our four main contributions: (i) we identify a structural mismatch between algorithm-centric regulation and socio-technical ML systems; (ii) we introduce the Cultural Expressions Discovery Circuit as a framework for understanding socio-technical ML systems and the ecosystem they inhabit; (iii) we apply the CEDC to contemporary regulatory practice, demonstrating how it exposes important blind spots in existing approaches to discovery governance; and (iv) we show how this model can be used to ameliorate regulatory gaps and inform future research objectives for the broader technical community.

## 2. Defining Discovery

In this paper, we adopt a broad definition of *discovery*, referring to it as *the general process by which an individual has a meaningful encounter with a content item that they were previously unfamiliar with*.

The development of recommendation algorithms on online platforms has often required formalizing the above notion of *meaningfully encountering unfamiliar items*. In practice, these algorithms analyze past engagement patterns on the service to learn individual preferences (Koren & Bell, 2015; Schedl et al., 2018). In this setting, an item is commonly treated as "discovered" once a measurable interaction–such as a stream, click, or view–has been observed and recorded (Mok et al., 2022; Tran et al., 2024; Moscati et al., 2025).

This technical abstraction oversimplifies discovery in ways that are consequential for both system design and regulation. Rather than a single measurable event, discovery is often abstracted to become an overarching concept encompassing several interrelated principles spanning technical, socio-technical, and policy-oriented literature: diversity, exposure, prominence, popularity bias, and serendipity. For example, ML research has documented how popularity bias can systematically advantage already-successful content, narrowing the range of items that users are exposed to and, thus, limiting their ability to discover niche or novel content (Salganik et al., 2024; Celma, 2010; Abdollahpouri et al., 2020). In parallel, policy-oriented scholarship has raised concerns about the cultural consequences of these dynamics, particularly in relation to the sustainability of local, independent, or underrepresented creative practices (Page & Dalla Riva, 2023; Turnbull et al., 2022). Meanwhile, research in human–computer interaction has examined how interface design, visualization, and navigational affordances influence exploratory behavior (Liang & Willemsen, 2021; Knees et al., 2020), while reinforcement learning has framed discovery within the exploration-exploitation trade-off, where systems must balance introducing novel content against reinforcing previously expressed preferences (Mehrotra et al., 2018). Finally, socio-technical research has explored the

underlying principles that govern the belief systems of the engineers designing curatorial systems (Born et al., 2021; McKelvey & Hunt, 2019; Seaver, 2022).

Crucially, the fragmentation of technical, socio-technical, and policy-oriented perspectives has made it difficult to translate high-level objectives–such as enabling diverse cultural exposure or promoting equitable representation–into concrete and actionable regulation. While these normative goals are broadly shared, existing approaches tend to target isolated components (e.g. ranking objectives, diversity metrics, or interface features) without addressing how their interactions shape users' overall discovery outcomes. This gap motivates the need for frameworks that can reason about discovery as an emergent property of the ecosystem surrounding ML-driven platforms, rather than a proxy measured at the level of individual algorithms.

## 3. The Cultural Expressions Discovery Circuit

We design the Cultural Expressions Discovery Circuit (CEDC) through an extensive literature review across various disciplines among which anthropology (cultural practices), economics (market incentives), sociology (social practices), computer science (algorithms), media studies (content and audiences), philosophy and law (fairness) play an important role. Named after the Culture Circuit presented by Du Gay et al., we design the CEDC to highlight the explicitly non-linear relationships between its individual components. Though we present them in an ordered way, we wish to stress that no fixed entry point is assumed. Instead, discovery unfolds through a series of interconnected cycles.

Ultimately, we present the CEDC to formalize why algorithm-centric regulation is not sufficient for governing the discovery of cultural expressions in online platforms. Due to space constraints, we briefly summarize the six components below and use them to justify our call to action (see Section 6). For more detailed descriptions of both our methodology and each individual component, please see Appendix C. We also note that the CEDC was recently adopted within a broader initiative led by the European Commission, which investigates how platform design and algorithmic systems influence exposure to culturally diverse content online (European Commission, 2024). The resulting policy report (Clarke et al., 2026) leverages the CEDC to structure proposed actions for promoting cultural diversity across music and related media (e.g. books and heritage-based multi-media).

### 3.1. Individual Components

**Engagements – The "Human" Inputs** capture the cultural, social, and individual conditions that shape whether and how users engage in discovery. Prior work shows that openness to novelty, content engagement practices, socioeconomic background, and technological literacy strongly influence discovery behavior, independently of platform design or recommendation quality (Peterson, 1992; Cunningham et al., 2007; Holzapfel et al., 2018; De Vries & Reeves, 2022). Since discovery is conditioned by users' cultural dispositions, resources, and capacities to engage with unfamiliar content, barriers to cultural participation, such as socioeconomic constraints, literacy gaps, or limited access to local artistic ecosystems, shape whether discovery occurs at all. From a circuit perspective, discovery-related governance extends beyond platform design into broader cultural infrastructures including education, cultural funding, and support for local cultural scenes. These contextual factors influence discovery indirectly by shaping users' readiness and ability to engage with new content. From a technical perspective, the contextual factors encompassed within Engagements become implicitly encoded in training data, user representations, and interaction signals. As a result, disparities in cultural capital, literacy, or access are absorbed into the model's latent representations of users.

**Mediations – The "UI/UX" Layer** refer to the media and interface environments through which content is presented to users. Interface layout, prominence, ordering, and navigational cues structure attention and shape exposure (Napoli, 2011; McKelvey & Hunt, 2019; Mazzoli, 2020; Diaz et al., 2020). In this context, visual design becomes a key site of analysis: discovery outcomes depend not only on what content is selected, but on how it is surfaced. Within the technical community interfaces constitute the operational boundary through which model outputs are presented for user behavior. Rankings, scores, or recommendations produced by ML systems only acquire meaning when engaged with by users—making design choices a critical mediating layer between predictive inference and users' discovery. Examining Mediations through the CEDC foregrounds the role of interface design, prominence patterns, and presentation or placement biases in shaping discovery trajectories.

**Facilitators – The "Engine"** encompass algorithmic, human, and hybrid curation systems that surface content to users. While recommender systems are often optimized for relevance or engagement, discovery-oriented objectives such as novelty, diversity, or long-term exposure frequently remain secondary, leading to well-documented biases related to popularity, language, gender, and genre (Ge et al., 2010; Ekstrand et al., 2018; Abdollahpouri et al., 2020; Salganik et al., 2024; Ferraro et al., 2021a). Facilitators instantiate particular optimization regimes that formalize discovery through measurable proxies such as clicks, watch time, or predicted relevance (He et al., 2023). These objectives embed normative assumptions about value and success, privileging content that aligns with dominant engagement patterns while systematically disadvantaging niche,

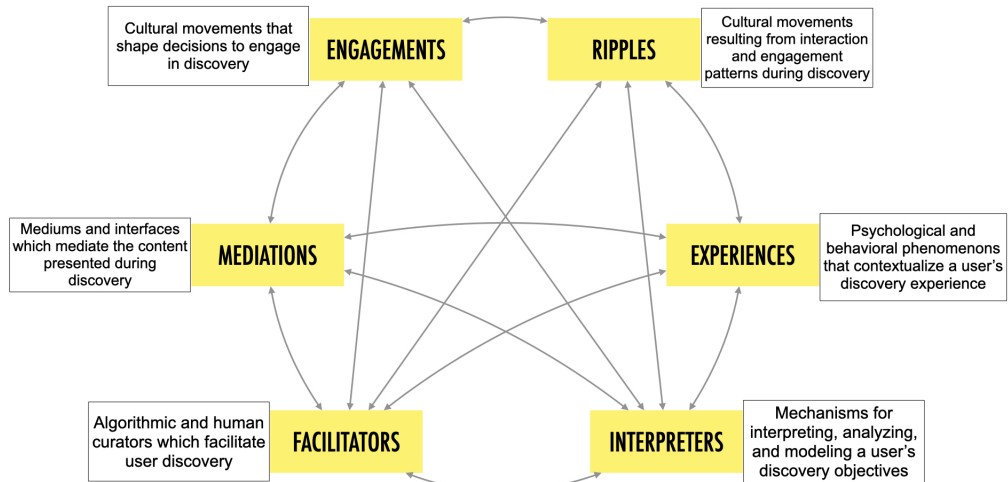

*Figure 1.* The Cultural Expressions Discovery Circuit, a cohesive framework for discovery within online platforms.

emergent, or culturally marginal forms of production (Born et al., 2021; Seaver, 2022). Analyzing Facilitators within the CEDC highlights how biases in inputs and outputs can accumulate, shaping long-term exposure patterns for both users and artists.

**Interpreters – The "Datafication" Layer** denote the mechanisms through which platforms model users by translating behavioral signals into representations of preference and intent. Aggregated user modeling can suppress dynamic or exploratory behaviors, contributing to homogenization effects and reinforcing normative assumptions about "typical" users (Born et al., 2021; Pedersen, 2020; Raff et al., 2020). In this way, Interpreters function as inductive biases that constrain how user behavior is interpreted and acted upon. Choices about feature construction, temporal aggregation, and representation learning determine which aspects of behavior are treated as signal versus noise, privileging stable, historically dominant patterns over transient, exploratory, or context-dependent engagements. Within the CEDC, Interpreters are understood as active components that shape future discovery by influencing curation and personalization strategies. This perspective draws attention to how assumptions embedded in user models affect not only recommendations, but the range of discovery experiences sustained over time.

**Experiences – The "Relationship"** capture users' subjective relationships with discovery systems, including their goals, trust, perceived agency, and listening contexts. Research shows that discovery intentions vary across time and situation, and that users' perceptions of algorithmic mediation shape how recommendations are interpreted and acted upon (Garcia-Gathright et al., 2018; Hosey et al., 2019; Freeman et al., 2023). From a circuit perspective,

Experiences influence how recommendations are perceived, feeding back into signals of Engagement and Facilitator updates (Caldwell Brown & Krause, 2016; Besseny, 2020). Changes in trust, perceived control, or interpretability can alter engagement behavior in ways that indirectly but persistently reshape learning dynamics and, thus, a model's latent representations (Guedes et al., 2023). This highlights the importance of considering user intents and perspectives on algorithmic systems when evaluating discovery outcomes.

**Ripples – The "Consequences"** represent downstream cultural and economic effects of discovery patterns. These include impacts on artist visibility, income distribution, genre survival, and even artistic form, as creators adapt production strategies to platform incentives (Aguiar, 2017; Morris, 2020; Hesmondhalgh et al., 2023). Importantly, these effects feed back into the system, reshaping catalogs, engagement signals, and future curation (Bauer et al., 2017). Crucially, these downstream effects shape the environment in which discovery systems learn, altering the distribution of content and interactions that models observe, effectively shifting the data landscape on which future recommendations are trained.

### 3.2. Circuit-Level Interactions and Emergent Outcomes

A defining feature of the Cultural Expressions Discovery Circuit is that discovery outcomes emerge not from individual components, but from their interactions. Several interaction patterns are particularly salient. Below, we comment on specific patterns in these internal and external cycles visualized in Figure 3, which are later used to contextualize our call to action.

First, interfaces and curation systems jointly shape exposure

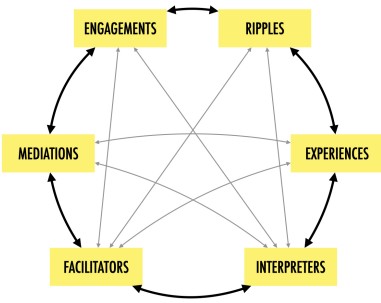

*(a) Overarching Discovery Flow*

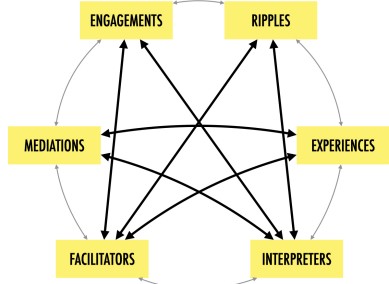

*(b) Internal Discovery Cycles*

*Figure 2.* The outer loop captures the *Overarching Discovery Flow*, the canonical discovery journey from Engagements to Ripples in the CEDC. The inner connections form the *Internal Discovery Cycles*, modeling flows between non-contiguous components.

(Mediations ↔ Facilitators). Interface design determines which recommendations are visible, salient, or easily actionable, while curation systems determine which items are eligible for exposure in the first place (Seaver, 2022). Empirical work shows that changes in prominence, ordering, or layout can significantly affect discovery and consumption even when underlying recommendation logic remains unchanged (Napoli, 2011; Mazzoli & Tambini, 2020; Knees et al., 2020; Liang & Willemsen, 2021). Conversely, advances in recommendation systems often prompt corresponding changes in interface design, reinforcing specific discovery pathways (Briand et al., 2024; Besseny, 2020).

Second, user modeling and experience are mutually reinforcing (Interpreters ↔ Experiences). Behavioral signals collected during listening sessions are used to infer preferences and optimize future curation, while users' perceptions of being accurately–or inaccurately–understood shape their trust in and interactions with platform recommendations (Garcia-Gathright et al., 2018; Freeman et al., 2022; 2023; Cunningham et al., 2024; Kizilcec, 2016). These dynamics mean that interpretive models are not passive representations of users, but actively shape how users interact with the system and what forms of discovery are sustained (Ytre-Arne & Moe, 2021; Siles et al., 2020).

Third, discovery patterns generate downstream cultural effects that feed back into the system (Facilitators ↔ Ripples). Exposure and consumption influence artist visibility, remuneration, and production strategies, which in turn affect catalog composition and future recommendation data (Aguiar, 2017; Hesmondhalgh et al., 2023; Morris, 2020). These feedback loops contribute to well-documented phenomena such as popularity bias and cultural homogenization, even when individual system components operate as designed (Ferraro et al., 2021b; Bauer et al., 2017; Karakayali et al., 2017).

Fourth, assumptions embedded in user models influence interface presentation through personalization and A/B testing (Garcia-Gathright et al., 2018; Morris et al., 2021;

| Legislation | Ctry. | Eng. | Med. | Fac. | Int. | Exp. | Rip. |
|---|---|---|---|---|---|---|---|
| *Legend* | | ✓ = Primary | | | ○ = Secondary | | |
| Digital Services Act (DSA) | EU | ✓ | ✓ | ✓ | | | |
| Digital Markets Act (DMA) | EU | ✓ | ✓ | | | | ○ |
| European Accessibility Act (EEA) | EU | ✓ | | | | ○ | |
| Audiovisual Media Services Directive (AVMSD) | EU | ✓ | ✓ | | | | ○ |
| European Media Freedom Act (EMFA) | EU | ✓ | | | | ✓ | |
| General Data Protection Regulation (GDPR) | EU | | | ○ | ✓ | | |
| EU Artificial Intelligence Act (AI Act) | EU | | | ✓ | ✓ | | |
| Online Safety Act (OSA) | UK | ✓ | ✓ | | | ✓ | |
| Algorithmic Transparency Recording Standard (ATRS) | UK | ○ | ✓ | ✓ | | | |
| Algorithmic Accountability Act | US | | | ✓ | ✓ | | |
| Online Streaming Act | CA | ✓ | ✓ | | | ○ | ✓ |
| Artificial Intelligence and Data Act (AIDA) | CA | | | ○ | ✓ | | |
| PIPEDA | CA | | | ○ | ✓ | | |
| Broadcasting Act modernization | CA | | | | | | ✓ |
| Provisions on Algorithmic Recommendations | CN | ✓ | ✓ | ✓ | ✓ | | |
| Deep Synthesis Provisions | CN | ✓ | ✓ | | | ○ | |
| Act on Improving Transparency and Fairness of Digital Platforms | JP | ✓ | ✓ | | | | |
| IT Rules (Intermediary Guidelines & Digital Media Ethics Code) | IN | ✓ | ○ | | | ✓ | |
| Digital Personal Data Protection Act (DPDP Act) | IN | | | ○ | ✓ | | |
| Personal Data Protection Act (PDPA) | SG | | | ○ | ✓ | | |
| Online Platform Fairness Act (proposed) | KR | ✓ | ✓ | | | | ○ |
| Act on Information & Communications Network | KR | | | ○ | ✓ | | |
| General Personal Data Protection Law (LGPD) | BR | | | ○ | ✓ | | |
| Marco Civil da Internet | BR | ✓ | ○ | | | | ✓ |

*Table 1.* Mapping selected regulatory measures to components of the Cultural Expressions Discovery Circuit (CEDC), illustrating the predominance of component-level interventions–particularly Facilitators and Interpreters. For further details, please see Appendix D.

*Notes:* Country abbreviations: EU (European Union), UK (United Kingdom), US (United States), CA (Canada), CN (China), JP (Japan), IN (India), SG (Singapore), KR (South Korea), BR (Brazil). Markers indicate whether a regulation targets a component *directly* (✓) or *indirectly* (○).

Hogan, 2015) (Mediations ↔ Interpreters), while user experiences feed back into consumption patterns that shape future data collection (Experiences ↔ Interpreters) (Gillespie, 2016; 2018). Furthermore, issues of accessibility and technological competence significantly affect users' interactions, expectations, and experiences with curatorial systems (Engagements ↔ Experiences) (Garcia-Gathright et al., 2018; Pedersen, 2020; Maasø & Spilker, 2022; Lee et al., 2023), highlighting how non-algorithmic components of the circuit are intrinsically linked to the outcomes of technical systems.

Finally, we also highlight the recursive relationship through which cultural contexts and individual motivations co-evolve through the dynamic interplay of discovery and identity formation (Ripples ↔ Engagements)(McCourt et al., 2016; Nowak, 2016). Such cycles illustrate why discovery outcomes are resistant to localized intervention: changes introduced at one point in the circuit are often mediated, amplified, or neutralized elsewhere. This structural property of the system is central to understanding why algorithm-centric regulation is insufficient for legislating discovery outcomes. In the remainder of the paper, we use the CEDC as an analytical lens to explain why regulation that targets algorithms or interfaces in isolation is structurally misaligned with outcome-oriented policy goals such as discoverability, diversity, and exposure.

## 4. Applying CEDC to Current Regulation

A growing body of regulation now governs how online platforms curate, rank, and present content. As a result, the retrieval, ranking, and recommendation systems developed by the ML community will increasingly operate under explicit regulatory constraints. Understanding this landscape is therefore essential for anticipating the objectives and evaluation criteria that will shape deployed systems. Table 1 maps major regulatory initiatives onto the Cultural Expressions Discovery Circuit (CEDC).

Across jurisdictions, regulation concentrates primarily on Facilitators and Interpreters, targeting algorithmic curation, ranking practices, and user profiling through transparency, auditing, and documentation requirements. These interventions reflect a shared assumption that improving algorithmic components will directly improve discovery outcomes. By contrast, relatively little regulation addresses Engagements, Experiences, or Ripples, which shape how users participate in discovery and how exposure patterns translate into longer-term cultural and economic effects.

This narrow focus on algorithmic components helps explain why reforms targeting algorithms alone often fail to produce sustained changes in discovery, diversity, or exposure dynamics. While technical interventions may improve fairness or transparency within individual systems, they leave unad-dressed the broader socio-technical conditions that govern how discovery outcomes emerge in practice. Addressing these limitations requires moving beyond algorithm-centric remedies toward broad system-level objectives. Doing so opens new directions for feedback-aware evaluation in recommender systems and for regulatory approaches that address exposure dynamics over time.

## 5. Why Algorithm-Centric Regulation Fails

Using the CEDC, we identify three structural limitations in the current body of legislation surrounding algorithmic systems. In doing so, we advocate for the importance of applying a socio-technical approach to improving discovery in online platforms and motivate our call to action.

### 5.1. Only Targeting Algorithmic Components

Most regulatory frameworks implicitly assume that accountability for platform outcomes can be meaningfully assigned to algorithmic components, in the form of data collection practices, recommender systems, or ranking models (Ulbricht & Yeung, 2024b). This framing aligns with a regulatory style that is predominantly compliance-driven. This means that platform governance is operationalized through transparency obligations, documentation requirements, and the threat of enforcement actions such as financial penalties for non-compliance (European Parliament and Council of the European Union, 2022; Parliament of Canada, 2023). We argue that while such approaches can be appropriate for mitigating harm, they may not align with policy objectives that are fundamentally constructive in nature, such as discoverability, cultural diversity, and sustained opportunities for new and niche creators. The CEDC enables a more holistic approach to regulatory design, that accounts for the components acting in the periphery of algorithmic systems. For example, Canada's implementation of the Online Streaming Act requires online streaming services with at least $25 million in annual revenues contribute 5% of those revenues to support cultural objectives (Canadian Radio-television and Telecommunications Commission (CRTC), 2024). Such novel mechanisms for public investment in local artistic infrastructures, highlight the importance of socio-technical components like Engagements and Ripples.

Similarly, within the technical community this framing translates into a tendency to center discovery objectives around individual models or datasets–through tools such as model cards, data documentation, or post-hoc audits–while treating downstream system behavior as external to the scope of evaluation (Winecoff & Bogen, 2024). Meanwhile, the internal cycles within the CEDC suggest complementary avenues for technical intervention. For example, we highlight how understanding engagement disparities through user-centered research (Engagements ↔ Facilitators), auditing

discovery experiences across populations (Interpreters ↔ Experiences), or examining how system design choices amplify or mitigate unequal outcomes over time (Mediations ↔ Ripples) can uncover important elements shaping users' discoveries.

## 5.2. Formalizing Cultural Objectives through Exclusively Technical Terminology

A second limitation of current governance and evaluation practices is the translation of outcome-level goals—such as discovery, exposure diversity, or fair access—into narrow technical proxies.

Within the CEDC, this gap is reflected in the distinction between Interpreters, which model user preferences from behavioral data, and Experiences, which describe users' goals and exploratory intent during interaction. While technical work has focused heavily on improving transparency, explainability, and profiling accuracy, far less attention has been paid to whether these representations support empowerment of exploration or diverse exposure in downstream use.

For the technical research community, this highlights an outcome–mechanism mismatch. Methods designed to satisfy technical objectives–such as transparency or debiasing–may not reliably translate into improved discovery when user behavior adapts over time and interacts with repeated exposure and feedback loops. Even transparent and well-calibrated models may still converge toward homogenized recommendations if they encode static assumptions about relevance or preference stability (Ananny & Crawford, 2018; Born et al., 2021). As a result, regulating or formalizing constraints for algorithms alone remains insufficient.

## 5.3. Applying Modular Thinking to the Entire Ecosystem

Finally, algorithm-centric regulation inherits a modular conception of systems that is common in engineering but poorly suited to encompassing socio-technical dynamics. Regulatory frameworks often assume that systems can be decomposed into separable components, each governed by targeted obligations (Whittaker et al., 2018). This assumption produces blind spots where harms emerge from interactions rather than from any single component. In the CEDC, these blind spots are especially apparent in the relationship between Facilitators and Ripples. Facilitators shape what content is surfaced and consumed, while Ripples capture downstream effects such as income distribution, genre viability, linguistic diversity, and creative adaptation (Ferraro et al., 2021a; Bauer et al., 2017; Prey, 2019; Aguiar et al., 2021). These effects are not direct outputs of recommendation algorithms, but cumulative consequences of repeated exposure patterns interacting with economic and cultural

incentives (Seaver, 2022; Mladenov et al., 2020). Similarly, we can see this gap in the intermediary connections between Interpreters and Ripples. For example, prior research shows that artists often come to reinterpret their own work in response to the genre labels and categorizations assigned by platforms (Prey, 2020b; Morris, 2020; Graham & Burnett, 2019). This illustrates how the normative assumptions embedded in algorithmic systems can shape cultural expressions.

From a technical design standpoint, this creates a blind spot: If algorithms are evaluated independently of the economic and cultural environments they shape, and downstream effects are treated as externalities rather than as integral system behaviors, algorithmic evaluation will systematically overlook emergent phenomena that arise precisely because systems are interconnected. Broad cultural objectives require circuit-level analysis, but modular analysis lacks the conceptual tools to engage with such dynamics.

## 6. Call to Action

This section illustrates the potential implications of the CEDC.

### 6.1. Research Agendas

6.1.1. For the Technical Research Community

The CEDC provides a circuit-level framing that helps technical researchers move beyond algorithm-centric approaches to discovery.

**1. Formalize discovery as a measurable objective rather than an implicit byproduct.** A primary challenge is that "discovery" is frequently invoked as a normative goal without a stable empirical formalization. In practice, it is fragmented across partially overlapping targets such as diversity, novelty, serendipity, popularity bias mitigation, catalog coverage, and long-tail exposure. This ambiguity makes discovery difficult to optimize for, audit, or compare across systems. Furthermore, the CEDC clarifies that discovery is not an intrinsic property of a recommender system, but a broad-based outcome shaped across the ML lifecycle (data selection, training objectives, ranking and playlist construction, online evaluation, deployment, and maintenance). This motivates a research agenda that treats discovery as an explicit evaluation target, analogous to how fairness objectives have become standardized within auditing pipelines.

**2. Translate circuit-level discovery goals into concrete research directions.** The CEDC surfaces many opportunities for future research directions that treat discovery as a dynamic, interdisciplinary phenomenon. For example, we can consider opportunities for feedback-aware evaluation that accounts for feedback effects under repeated exposure, rather than relying solely on one-shot relevance or diversity

metrics (Chaney et al., 2018; Jiang et al., 2019; Schnabel et al., 2016). Similarly, we can consider how optimization objectives should evolve beyond single-stakeholder utility to consider the interplay between user satisfaction, creator opportunity, and catalog evolution (Mladenov et al., 2020; Mehrotra, 2021). Our framework also highlights the importance of interface design as an integral part of recommendation systems, suggesting that prominence and layout should be integrated into evaluation of recommendations rather than fixed presentation layers (Petridis et al., 2022). Finally, the circuit perspective encourages longitudinal analysis of discovery trajectories, where shifts in exploration behavior, novelty tolerance, and exposure concentration are examined over time rather than inferred from isolated interactions.

### 6.1.2. For Regulators

The CEDC offers regulators a circuit-level lens for designing interventions that match the socio-technical mechanisms through which discovery outcomes are produced.

**1. Build monitoring capacity and institutional infrastructure.** Because online platforms and their ecosystems evolve rapidly, effective governance requires longitudinal monitoring rather than one-off compliance checks. Regulators can support regular reporting and indicators (e.g., exposure diversity measures and distributional outcomes across creator groups).

**2. Use enabling instruments to strengthen engagements and ecosystem-level resilience.** The CEDC highlights that discovery outcomes depend not only on platform curation but also on upstream conditions for cultural participation and downstream incentives for production. Regulators can therefore complement compliance-driven governance with enabling interventions that strengthen Engagements and Ripples, such as investments in digital and media literacy, support for local artist infrastructures, and targeted funding for underrepresented scenes. More broadly, policymakers can fund experimentation through "sandbox" environments in which platforms, cultural institutions, and researchers test discovery interventions against exposure-diversity objectives in controlled settings.

### 6.2. Concrete Example: Applying CEDC to Music Streaming Contexts

We also provide a domain-specific example to clearly illustrates how the framework can be operationalized. For a longer discussion of how the framework may be applicable beyond music or media recommendation settings, please see Appendix E.

### Engagements:

- *Technical goal*: Model and account for heterogeneity in user readiness for discovery (e.g., differences in cultural participation as a function of infrastructure and resources).
- *Regulatory goal*: Support cultural participation through investments in local music ecosystems (e.g., community programs, education, or funding for local scenes).

### Mediations:

- *Technical goal*: Optimize interface design for balanced exposure (e.g., prominence-aware ranking and layout evaluation).
- *Regulatory goal*: Ensure fair and non-manipulative presentation of content (e.g., constraints on prominence, avoidance of dark patterns).

### Facilitators:

- *Technical goal*: Develop recommendation algorithms that balance relevance with diversity and long-term exposure.
- *Regulatory goal*: Impose accountability on ranking systems (e.g., transparency, auditing, or diversity-related obligations).

### Interpreters:

- *Technical goal*: Learn richer user representations that capture exploratory intent and dynamic preferences.
- *Regulatory goal*: Govern user modeling practices (e.g., limits on profiling, requirements for user control and interpretability).

### Experiences:

- *Technical goal*: Design systems that support user agency and exploration (e.g., controllable or interactive recommendation modes).
- *Regulatory goal*: Ensure user rights to transparency and control over recommendation systems (e.g., ability to adjust or opt out of personalization).

### Ripples:

- *Technical goal*: Model and evaluate long-term ecosystem effects (e.g., impact on content supply, creator incentives, and distributional outcomes).
- *Regulatory goal*: Address downstream cultural and economic effects (e.g., requirements for data access, funding mechanisms, or monitoring of exposure outcomes).

Together, these interventions illustrate how discovery outcomes emerge from coordinated changes across components, rather than from algorithmic adjustments alone.

## 7. Alternative Views

This section addresses several plausible counterarguments to our critique of algorithm-centric regulation. Due to space

limitations, we provide additional discussion points in Appendix B.

**1. Maybe algorithmic intervention is sufficient.**

A dominant perspective in platform governance debates holds that since algorithms act as the core infrastructure through which visibility, participation, and attention are distributed, their regulation is the most effective mechanism for addressing issues in online content discovery.

This perspective is attractive because it is technically tractable, measurable, and aligns with existing regulatory tools such as transparency, auditing, and fairness constraints. Under this view, modifying algorithmic objectives (e.g., to promote diversity or mitigate popularity bias) is enough to enact meaningful changes in the discovery practices of users on a particular platform. For example, modifying ranking (Abdollahpouri et al., 2020; Jaenich et al., 2024), diversifying recommendations (Baracskay et al., 2022; Porcaro et al., 2021), or adjusting discovery mechanisms can reshape incentives for creators (Briand et al., 2024), influence audience engagement (Briand et al., 2024; Mehrotra et al., 2018), and alter the conditions under which local organizations and cultural producers operate (Born et al., 2021; Ferraro et al., 2021a). Accordingly, changes to algorithmic design are often viewed as capable of producing the desirable ecosystem-wide effects that would facilitate better discovery.

We agree that algorithmic design is a powerful lever for improving discovery patterns in online platform. However, our argument is that algorithmic interventions primarily operate on observed interaction data and affect upstream and downstream factors only indirectly. For instance, as mentioned in earlier portions of this text, re-ranking content to promote a particular, under-represented group of artists does not address structural constraints such as limited resources (Born et al., 2021; Prey, 2020b). Similarly, it will not have significant immediate consequences on the market environment in which a platform competes (Hracs & Webster, 2021; Drott, 2023). Our position is therefore not to downplay the importance of algorithms, but to emphasize that they address only part of the system. Achieving sustained changes in content production or representation likely requires complementary interventions that also act on the broader socio-technical ecosystem and underlying data-generating process.

**3. Maybe "discovery" is too subjective to quantify or regulate.** As discussed in Section 6, we fully acknowledge that emergent cultural behaviors cannot be fully captured by formal models and thus even defining "discovery" remains an open question within the scientific community. Our goal in designing the CEDC is to account for this ambiguity by providing an explicitly interdisciplinary approach which combines granular socio-technical methods (e.g., HCI, anthropology, and sociology, which are particularly well-suited to capturing edge cases, situated practices, and non-formalizable behaviors) with aggregate empirical analyses (e.g., large-scale interaction data and economic indicators). The usefulness of the CEDC lies in providing a unified vocabulary that situates these different forms of evidence within a shared system, making explicit how insights from one component (e.g., user behavior or creator incentives) propagate through the ecosystem and affect algorithmic outcomes. In this sense, the CEDC does not require fully specified models of cultural behavior, but instead enables structured interdisciplinary reasoning in settings where such behaviors are inherently difficult to formalize.

## 8. Conclusion

This paper has argued that outcome-oriented regulation of algorithmically-driven platforms is structurally misaligned with the way such systems actually produce effects. The Cultural Expressions Discovery Circuit (CEDC) provides a compact way to make this misalignment visible. By synthesizing insights from machine learning, human–computer interaction, media studies, cultural economics, and policy research, the CEDC translates a fragmented literature on discovery into a single framework. We illustrate that discovery emerges as a property of the entire ecosystem rather than of any individual technical artifact. Cultural dispositions and structural conditions shape users' willingness and capacity to engage in discovery; interface design governs visibility and attention; curation systems prioritize certain forms of relevance over others; interpretive models encode assumptions about users; experiences mediate trust, agency, and intent within user groups; and downstream effects reshape cultural production and economic incentives. These dynamics unfold over time, making discovery outcomes resistant to governance over isolated components. More broadly, this work highlights the need for ML design and governance to reconsider its unit of analysis. We propose several future directions for both technical researchers and regulatory bodies: advocating for formalizing discovery as a concrete technical objective and then harnessing this objective to design monitoring techniques at the institutional level. Broader, interdisciplinary perspectives, such as the CEDC, provide one way to bridge this gap, aligning regulatory attention with the realities of contemporary ML systems without reducing the scope of governance to the properties of individual components.

## Acknowledgments

This publication draws on research conducted as part of the study titled Study on the discoverability of diverse European cultural content in the digital environment, commissioned by the European Commission's Directorate-General for Education, Youth, Sport and Culture (DG EAC) under the EU Work Plan for Culture 2023–2026. The authors gratefully acknowledge the support of DG EAC and the wider study consortium. The views expressed in this publication are those of the authors and do not reflect the position of the European Commission.

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

# A. Related Work: Socio-Technical Perspectives on Algorithmic Systems

A substantial body of interdisciplinary scholarship has challenged the treatment of algorithms as isolated technical artifacts, instead emphasizing their embeddedness within broader socio-technical systems. Our work builds on this tradition while extending it toward a circuit-level model explicitly designed to diagnose governance failures around discovery and exposure outcomes.

## A.1. Algorithms as socio-technical assemblages.

Early and influential work in science and technology studies (STS) and media studies has argued that algorithms cannot be meaningfully understood apart from the institutional, cultural, and practical contexts in which they operate (Ananny, 2016; Bonini & Magaudda, 2024; Dolata et al., 2022; Gillespie, 2018; Pasquale, 2015). Ananny's notion of networked information algorithms conceptualizes algorithms as assemblages of computational code, design assumptions, institutional logics, standards, and user practices rather than discrete mathematical objects. This framing highlights that algorithms become socially and ethically consequential only through their relationships with users, organizations, and data flows, and that their influence often remains invisible until failure occurs. Related work similarly dissolves the boundary between "the algorithm" and its surrounding infrastructure, emphasizing that algorithmic power emerges through distributed socio-technical arrangements rather than from code alone.

## A.2. Limits of transparency and algorithmic accountability.

A second line of work interrogates the assumption that algorithmic accountability can be achieved primarily through transparency (Johnson, 2021; Loring, 2025). Scholars have shown that transparency has inherent temporal, epistemic, and practical limits: knowing how a system works does not necessarily confer control, nor does disclosure reliably translate into meaningful oversight (Ulbricht & Yeung, 2024b). Ananny & Crawford argue that transparency regimes often abstract away relational and dynamic aspects of algorithmic systems, while others note that calls to "open the black box" may paradoxically reinforce perceptions of algorithmic power (Pasquale, 2015). Empirical and theoretical work in this area demonstrates that algorithmic effects unfold over time through feedback loops, adaptation, and situated use, complicating efforts to govern outcomes via static disclosures, documentation, or audits alone (Ulbricht & Yeung, 2024b).

## A.3. Normativity, power, and emergent effects.

A third strand examines how algorithms participate in the production of norms, values, and power relations without acting as autonomous moral agents (Morris & Powers, 2015; Hesmondhalgh et al., 2023; McKelvey & Hunt, 2019; Seaver, 2022). Rather than locating responsibility or bias in algorithms, this literature emphasizes how normative effects arise from the interaction between computational systems, human practices, economic incentives, and cultural expectations. Ethnographic and historical studies of algorithmic systems show that ethical and political consequences often emerge indirectly, through repeated use, institutional uptake, and user adaptation (Gillespie, 2018; 2016). As a result, straightforward attributions of outcomes to "the algorithm" frequently fail empirical scrutiny.

## A.4. Positioning the CEDC

The Cultural Expressions Discovery Circuit (CEDC) builds directly on these insights while addressing a gap in existing work. Whereas prior scholarship has primarily focused on understanding or critiquing algorithms as socio-technical assemblages, the CEDC formalizes these insights into a diagnostic, circuit-level model tailored to governance questions. In particular, it makes explicit how discovery outcomes emerge from interactions among algorithms, interfaces, user modeling, engagement practices, and downstream cultural and economic feedback loops. By structuring these relationships into a circuit, the CEDC provides a concrete analytical tool for explaining why algorithm-centric regulation fails and for identifying leverage points where interventions can meaningfully affect exposure and discovery outcomes.

# B. Additional Alternative Views

**1. Maybe component-wise regulation is not the problem, enforcement is.** We do not deny the importance of enforcement. However, meaningful enforcement presupposes a regulatory basis that specifies what is being governed and why. The CEDC highlights that even under good-faith enforcement, component-level transparency and accountability instruments can have structural limits when regulatory objectives are circuit-level outcomes such as discovery. For example, while the Digital Services Act introduces transparency-related requirements for recommender systems, it does not straightforwardly encode how algorithmic interventions should be evaluated against downstream outcomes such as the consumption and circulation of creative content. Similarly, concerns about "the algorithm" in music streaming debates often obscure the broader ecosystem dynamics that shape both consumption and production (Hesmondhalgh et al., 2023). We therefore argue for constructive legislative actions that engage the discovery ecosystem as a socio-technical circuit rather than as an isolated problem of

model accountability.

**2. Is circuit-level regulation realistic, or does it become "regulate everything"?** Circuit-level regulation aims to be more precise rather than more expansive, aligning regulatory instruments with the socio-technical mechanisms that actually produce the outcomes of interest. In practice, circuit-level analysis can reduce regulatory burden by revealing that certain objectives cannot be achieved through algorithmic constraints alone, while others can be addressed through non-algorithmic instruments such as investment obligations or monitoring requirements. For instance, if the policy objective is increased exposure of local or minority-language music, circuit-level analysis may reveal that modifying ranking models has limited effect relative to interventions at the level of interface prominence or upstream production incentives.

# C. Cultural Expressions Discovery Circuit (In Detail)

Online discovery has been examined from a wide range of technical, cultural, behavioral, and policy standpoints. Collectively, this scholarship contributes to our understanding of the stakeholders, mechanisms, characteristics, and consequences of platformization on the discovery of cultural expressions.

However, the socio-technical literature is uneven across media. In our review, music was substantially over-represented relative to other forms of cultural expressions such as dance, cultural heritage, visual arts, and literature. As a result, the CEDC is primarily grounded in music discovery as its empirical and conceptual anchor. Our goal was not to produce an exhaustive systematic review of all cultural domains, but rather to synthesize recurring socio-technical mechanisms associated with platform discovery into a compact, governance-oriented model. We therefore acknowledge that some mechanisms identified in music streaming may not transfer directly to other contexts, where institutional structures, interaction modalities, and cultural practices differ.

Nonetheless, the governance problem motivating the CEDC is not unique to music. In many algorithmically mediated environments, outcomes such as exposure diversity and opportunity are emergent properties of interacting components, shaped by multi-stakeholder adaptation and feedback loops between exposure, production incentives, and downstream market structure. For this reason, we present the CEDC not as a universal blueprint, but as a transferable diagnostic template: a framework that can be adapted to other domains by redefining domain-specific components and interactions while preserving the core insight that circuit-level objectives require circuit-level analysis.

## C.1. Contextualizing Discovery

### C.1.1. A CLEAR DEFINITION

As mentioned in the main body of this work, we adopt a broad definition of discovery, referring to it as *the general process by which an individual has a meaningful encounter with an item that they were previously unfamiliar with*. This formulation is shaped by a large body of research that is centered around music discovery and our goal in keeping such an abstract notion was to enable the CEDC to be as applicable to other domains as possible.

Admittedly, this view oversimplifies the nuances of music discovery. First, the absence of interaction may indicate that the user has not listened yet to the item *on the service in question*, rather than genuine unfamiliarity. Second, listening contexts can vary considerably: lean-back, passive sessions involve reduced attention to the music, questioning the *meaningfulness* of such interactions when compared to more active, intentional listening experiences (Freeman et al., 2023). Third, prior work shows that discovery patterns are dynamic; *repeated exposure* is sometimes required to fully internalize a musical piece (Mehrotra, 2021; Sguerra et al., 2022). Consequently, richer formalizations of music discovery, e.g., requiring multiple streams or cross-platform signals, can be valuable for certain applications (Nowak, 2016). Importantly, however, our CEDC remains general and compatible with any formalization preserving the idea of a process through which individuals meaningfully encounter unfamiliar music.

### C.1.2. RELATED WORK AND CONCEPTS

Music discovery has been extensively studied in the existing research on online music consumption and recommendation. A noticeable portion of the technical literature originates from music streaming services themselves. For example, companies like Spotify and Deezer have designed several recommendation algorithms to support music discovery, including the *Discover Weekly*, *Fresh Finds* (Jacobson et al., 2016), *Track Mix* (Bendada et al., 2023), and *New Releases for You* features (Briand et al., 2024). Beyond algorithmic developments, various exploratory studies have also examined discovery using music streaming data. (Garcia-Gathright et al., 2018) investigated user satisfaction in discovery-oriented recommendations, noting that even a single well-liked song can make a discovery session feel successful. (Mok et al., 2022) analyzed *exploration* behaviors and dynamics, showing that users often engage in discovery in seasonal cycles. (Sguerra et al., 2022) found that interest in newly discovered music typically follows an inverted-U pattern: rising with repeated plays, peaking, and then declining, in a form that is consistent with the *mere exposure effect* in psychology (Zajonc, 1968). (Moscati et al., 2025) examined discovery patterns linked to different

user needs, showing that listeners who declare interest in unfamiliar music tend to explore more and display more diverse preferences.

Within the broader academic community, discussions of music discovery are typically fragmented and framed through related concepts with differing perspectives. Among these concepts, the most closely related is *discoverability*, which shifts the focus from the user to the online platform (Porcaro et al., 2024). Discoverability refers to how easily a piece of music can be found and the mechanisms platforms use to ensure its proper *exposure* to users. Enhancing discoverability not only supports discovery for users but also benefits artists by increasing their visibility and chances of reaching new audiences (Aguiar, 2017; Ferraro et al., 2021b). This is particularly relevant in the context of *platformization*, increasing the influence of platforms in shaping cultural production and circulation (Poell et al., 2019). Platformization enables multi-sided interactions, e.g., between users, or between users and creators (Nieborg & Poell, 2018). It reshapes musical presentation, e.g., by prioritizing songs or playlists over albums (Hiller & Walter, 2017). It also introduces challenges, particularly due to the limited transparency of algorithms and the conflicting interests of the actors who design, operate, and regulate them (Gillespie, 2018; Ranaivoson, 2010). Two related concepts are *prioritization* and *prominence*. Prioritization refers to the logics that determine which content is surfaced to users. Prominence, an outcome of prioritization, concerns how content is positioned within interfaces. Such decisions can facilitate or hinder music discovery (Mazzoli, 2020).

Music discovery is also linked to *diversity*, which considers the variety of content produced, made available to, and consumed by users (Ranaivoson, 2010; Mazzoli, 2020; Duricic et al., 2021; Porcaro et al., 2021). In the policy domain, significant efforts have been made to assess the impact of digital technologies on cultural diversity (Octavio, 2016; Pasikowska-Schnass, 2021), while technical research has proposed diversity metrics to train or evaluate recommender systems (Ge et al., 2010; Vargas et al., 2014; Schedl & Hauger, 2015; Baracskay et al., 2022). *Exposure diversity*, in particular, refers to the visibility allocated to different content and the feedback loop between what is presented to users and what they engage with (Diaz et al., 2020; Mazzoli & Tambini, 2020; Towse & Navarrete Hernández, 2020; Ranaivoson & Domazetovikj, 2023). This concept is tied to discovery, as engagement with a variety of music depends on access and awareness of it (Napoli, 2011). The concept of *serendipity* further enriches the picture. It describes the experience of encountering unfamiliar content unintentionally, yet finding it fortuitous and rewarding. Research in human-computer interaction has shown that music discovery carries a serendipitous quality, with timing playing a crucial role in shaping experiences (Hosey et al., 2019; Binst

et al., 2025). Finally, music discovery has been examined through the lens of *fairness*, amid growing concerns that platform curation and recommendation may fail to promote a diverse cultural landscape and unequally affect the visibility of certain artists or genres, ultimately limiting their chances of being discovered (Jin et al., 2023; Salganik et al., 2024; Burke et al., 2025).

## C.2. Methodology

Our goal was not to produce an exhaustive systematic review, but rather to synthesize recurring socio-technical mechanisms associated with content discovery into a compact governance-oriented model.

### C.2.1. DESK REVIEW AND SOURCE COLLECTION

We conducted a desk review spanning technical and socio-technical scholarship related to content discovery and algorithmic mediation. On the technical side, we surveyed major computer science and machine learning venues commonly associated with recommender systems, information retrieval, user modeling, and web-scale platforms, including RecSys, SIGIR, KDD, UMAP, CIKM, WWW, WSDM, and ICML, among others. We searched for publications from 2008 onward using keyword queries including (but not limited to): *discovery*, *music*, *playlist*, *music recommendation*, *books*, *video recommendation*, *recommendation interfaces*, and *ranking*, *diversity*, *filter bubbles*, *popularity bias*, and *serendipity*.

To complement this technical corpus with foundational socio-technical perspectives, we polled a broad group of researchers across adjacent disciplines—including law, regulatory policy, sociology, anthropology, and economics—to identify seminal works relevant to governance, cultural diversity, market dynamics, and platform mediation. We added these recommendations to the technical corpus to form a unified reading repository.

In total, this process produced a repository of over *100* works, spanning empirical studies, theoretical frameworks, policy analyses, and technical system descriptions.

### C.2.2. QUALITATIVE SYNTHESIS AND CONCEPT TAGGING

We reviewed the collected materials and performed a qualitative synthesis to identify recurring concepts that appeared across technical and socio-technical accounts of music discovery. During reading, we tagged passages that described (i) system mechanisms and design choices, (ii) user-facing discovery experiences, (iii) intermediaries and promotional practices, and (iv) downstream ecosystem outcomes (e.g., shifts in creator strategy, income concentration, or genre viability).

We then iteratively consolidated these tags into higher-level categories based on conceptual similarity and repeated co-occurrence across sources. This procedure was designed to emphasize cross-domain regularities (e.g., mechanisms appearing in both recommender systems literature and cultural-policy debates), rather than domain-specific details tied to particular platforms or datasets.

### C.2.3. DERIVING THE COMPONENTS

The final components were derived through iterative abstraction. Specifically, we grouped recurring concepts into six higher-level components that are intended to be interpretable across both technical and governance contexts. These components were named to reflect the underlying abstraction they capture rather than any single implementation detail. We refined the component definitions by repeatedly checking whether each component (i) represented a distinct part of the discovery process, (ii) captured a meaningful locus of intervention or measurement, and (iii) could be related to other components via plausible feedback pathways.

This synthesis yielded the six components presented in the main paper. Importantly, the CEDC is intended as a diagnostic governance map rather than a complete causal model; its purpose is to clarify where discovery outcomes may emerge from interactions across components and where isolated, module-level interventions may fail to translate into system-level objectives.

### C.3. Individual Components

### C.3.1. ENGAGEMENTS

Engagements encompass the cultural movements and factors that shape a user's decision to engage in discovery. Indeed, various forces influence a user's desire, ability, and opportunity to embark on a discovery journey. Research highlights how a user's demographics and practices shape their openness, ability, and willingness to discover new content (De Vries & Reeves, 2022; Lee et al., 2023). These include personal motivations (Garcia-Gathright et al., 2018), mindsets (Hosey et al., 2019), everyday practices (Cunningham et al., 2007), and levels of technological literacy (Holzapfel et al., 2018; Johnson et al., 2023). More broadly, sociological literature links socioeconomic factors, such as parental education, to users' openness to diversity in cultural consumption (Peterson, 1992; Sullivan & Katz-Gerro, 2006). Cultural environments also shape patterns of music production and circulation, which in turn influence user attitudes toward discovery (Nowak, 2016). For example, local music scenes, or environments where musicians gather in person, can affect the openness and listening habits of those living in the area (Straw, 2018).

### C.3.2. MEDIATIONS

Mediations encompass the media ecosystems through which information is presented to users engaging with music on online platforms (Besseny, 2020; Morris & Powers, 2015). Rather than focusing on what is presented, Mediations concern how content reaches users' senses, emphasizing the design choices behind its organization. These decisions vary in granularity, from devices and data protocols to font, color, and navigational cues within interfaces (McKelvey & Hunt, 2019).

Besseny (2020) introduces the concept of *wayfinders*, implicit pathways that direct user attention toward certain content and influence discovery. Similarly, McKelvey & Hunt (2019) describe *surrounds* as broader technological ecosystems (apps, dashboards, digital assistants) that shape interaction through heuristics of size, position, and contextual placement. Expanding on these ideas, Maasø & Spilker (2022) identify six techniques used by *hybrid gatekeeping mechanisms* to steer or nudge users by foregrounding some content and making others less accessible.

A substantial body of research examines the link between capturing user attention and consumption outcomes (Garcia-Gathright et al., 2018; He et al., 2023; Taramigkou et al., 2013; Knees et al., 2020). For example, information retrieval studies provide models that predict attention based on the order in which recommendations are presented (Diaz et al., 2020; Petridis et al., 2022). Fairness research highlights *exposure bias*, which measures disparities in visibility across item categories (Diaz et al., 2020; Ferraro et al., 2021a; Salganik et al., 2024). Empirical work further shows that interface placement directly impacts exposure, discovery, and consumption, such as the financial benefits of songs placed on popular playlists (Aguiar, 2017). Collectively, these findings emphasize how the positioning and presentation of information within Mediations significantly shape user discovery journeys.

### C.3.3. FACILITATORS

Facilitators encompass the full spectrum of content curation methods on platforms, including algorithmic, human, and hybrid (or algo-torial) practices (Bonini & Gandini, 2019; Maasø & Spilker, 2022). (Seaver, 2022) traces the origins of these practices, highlighting how the vast expansion of available content created user overwhelm (Schwartz, 2004), which Facilitators were designed to address. This challenge has driven innovation in algorithmic recommender systems (Xia et al., 2024). Given the competitive and lucrative nature of music streaming, scholars note that these systems are central to user satisfaction, retention, and platform loyalty (Drott, 2023; Gillespie, 2018).

As Facilitators gain prominence, research has increasingly

examined their power to shape discovery practices through curation (Gillespie, 2018; Bonini & Gandini, 2019; Born et al., 2021; Seaver, 2022). This is closely tied to the concept of discoverability: music that is never exposed or recommended can hardly be discovered (Bauer et al., 2017; Diaz et al., 2020). Crucially, however, discovery is sometimes not the primary goal of curatorial systems. Algorithms are typically optimized for relevance, understood as the likelihood that a user will *like* a recommended track (Ge et al., 2010). This distinction matters because discovery involves risks: unfamiliar music may not immediately appeal to users (Sguerra et al., 2022).

### C.3.4. INTERPRETERS

Interpreters encompass the various mechanisms used to analyze user behavior and preferences. Users engage with music on online platforms through explicit or implicit actions (e.g., skipping, listening, liking), which serve as proxies for item relevance (Cunningham et al., 2024). Efforts by platforms to thoroughly model and predict user preferences directly influence how curatorial systems are designed and refined (Gillespie, 2014; 2018; Drott, 2018; Born et al., 2021).

Importantly, this aspect introduces biases into the curation assemblage. As (Born et al., 2021) explain, the process of datafication assumes that tastes "*evolve according to a universal logic derived from an analysis of aggregated behavior of millions of listeners,*" often excluding unpredictable or dynamically evolving preferences. Similarly, Pedersen (2020) argues that relying on aggregated patterns to infer preferences leads to homogenization effects, resulting in a "*convergence of listening practices*" on platforms. Raff et al. (2020) note that "*different people might want to discover music in dissimilar ways while devoting different amounts of control,*" yet many platforms design their algorithms around an imagined "*normative*" user. Research even suggests that measuring engagement through implicit signals like listening times and skips can lead platforms to favor forms of music discovery that maximize passive listening over active discovery (Drott, 2018).

### C.3.5. EXPERIENCES

Experiences encompass the psychological phenomena that shape a user's engagement with platforms. More concretely, they reflect the varying intentions or personas (Trocchia et al., 2011; Lee & Price, 2015) that users adopt during music discovery. For instance, Trocchia et al. (2011) identified 11 types of experiences users seek, such as memory triggering, inspiration, and self-identification. Lee & Price (2015) describe several listener personas, including the *Active Curator*, who actively seeks novel music, and the *Nonbeliever*, who rejects curation and prefers only self-selected content.

Overall, users may approach discovery sessions with different goals or mindsets (Moscati et al., 2025).

Unlike Engagements, which remain relatively static or evolve slowly, Experiences are highly dynamic. For example, (McKelvey & Hunt, 2019) show that the same user can pursue different objectives at different times on a platform. Similarly, research highlights shifts in discovery intent as users move between active discovery, where they seek music for focused listening, and passive discovery, where they rely on a curatorial system to surface content (Garcia-Gathright et al., 2018; Raff et al., 2020; Mok et al., 2022). Crucially, user experiences are also shaped by their relationship with algorithms. Freeman et al. (2023) relate these socio-technical relationships to human dynamics such as trust, betrayal, and intimacy.

### C.3.6. RIPPLES

Ripples encompass the broader cultural practices and conditions that arise from, or are influenced by, widespread music discovery on platforms. For example, improved discovery facilitation has positively impacted the diversity of music consumption (Aguiar, 2017; Datta et al., 2017), as well as the representation of languages (Rioux, 2020), ideas, and perspectives (Helberger, 2018; Bello & Garcia, 2021).

## C.4. Connections

This section emphasizes the articulations between the CEDC's seemingly discrete components. While the previous section introduced each component as an analytically distinct part of the discovery ecosystem, music discovery on online platforms does not unfold through isolated modules. Rather, it emerges through the *intersections* between cultural motivations, interface design, curatorial logic, user modeling practices, lived experience, and downstream cultural and economic conditions.

Our proposed CEDC offers a unified framework that models music discovery as a dynamic process unfolding through entangled and often complementary pathways. Figure 3 illustrates two perspectives on this interdependence. The first is the *Overarching Discovery Flow*, which traces the canonical discovery journey around the circuit. The second is the *Internal Discovery Cycles*, which capture shorter feedback loops between non-contiguous components that often operate simultaneously alongside the outer flow.

### C.4.1. OVERARCHING DISCOVERY FLOW

To motivate the CEDC outer loop, we draw on the implicit models of music engagement of (Seaver, 2022) in his anthropological study of recommendation engineers at music streaming services. For pedagogical purposes, we present these bidirectional flows via the illustrative example of a

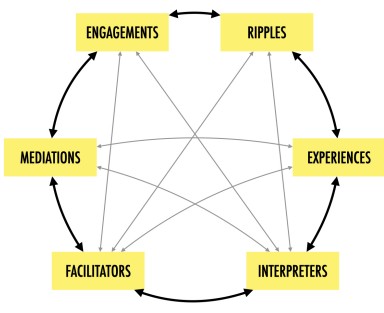
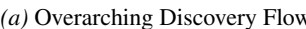

*(a)* Overarching Discovery Flow

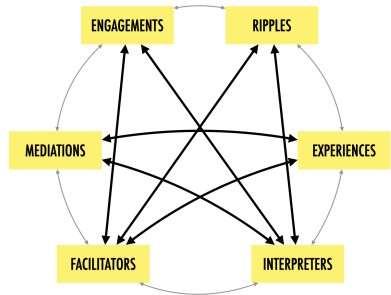

*(b)* Internal Discovery Cycles

*Figure 3.* The outer loop captures the *Overarching Discovery Flow*, the canonical discovery journey from Engagements to Ripples in the CEDC. The inner connections form the *Internal Discovery Cycles*, modeling flows between non-contiguous components.

generic user named *Alex*.

**Engagements ↔ Mediations**: Influenced by the factors outlined in Engagements—such as technical literacy, openness to novelty, and musical expertise—Alex chooses to discover new music. She navigates to her preferred streaming service, guided by personal preferences and familiarity. User experience design plays a key role here, aiming to attract the "*right kind of user*" (Prey, 2019; Seaver, 2022). Through deliberate design choices such as interface layout, content prominence, and personalized playlists like *Discover Weekly* (Jacobson et al., 2016), the platform subtly steers Alex toward specific forms of engagement. This illustrates the interplay between cultural motivations for discovery and the affordances of digital music technologies.

**Mediations ↔ Facilitators**: Alex's interactions with the platform are channeled through various Mediations that present her with content to engage with. These interactions are then processed by Facilitators, and later by Interpreters, to generate personalized discovery experiences calibrated to her tastes (Salganik et al., 2024). This process exemplifies the interdependence between the Mediation and Facilitator components. Interface design, shaped by Mediations, encourages users to engage with curatorial systems governed by Facilitators (McKelvey & Hunt, 2019; Liang & Willemsen, 2021; Petridis et al., 2022; Briand et al., 2024). Conversely, the technical capacities of Facilitators influence how content is presented and accessed. For instance, innovations in algorithmic design increasingly guide platforms to prompt both active and passive users toward engagement (Garcia-Gathright et al., 2018; Raff et al., 2020; Mok et al., 2022). These dynamics illustrate how digital media systems embed specific forms of expertise and assumptions, shaping user interaction and curatorial decisions.

**Faciltators ↔ Interpreters**: As Alex engages with a playlist, her interactions are analyzed by an Interpreter, transformed into algorithmic representations of her pref-

erences, and stored for future use (Garcia-Gathright et al., 2018). For instance, skipping a track may be interpreted as disinterest, causing similar songs to be de-prioritized in later recommendations (Prey, 2017). These user modeling practices enable Interpreters to infer intent and preferences, which in turn inform the objectives of algorithmic Facilitators (Mehrotra et al., 2018). Meanwhile, many of the behavioral signals used to model users originate from their responses to recommended content selected by curatorial systems, such as recommendation algorithms (Xia et al., 2024) or human curators (Aguiar et al., 2021). This reciprocal relationship between Interpreters and Facilitators reveals how music programming both depends on and contributes to data analytics and evaluative metrics, ultimately shaping future curatorial strategies.

**Interpreters ↔ Experiences**: Alex's satisfaction with a personalized recommendation subtly shapes her relationship with the platform. When an Interpreter accurately assesses her preferences, it fosters a sense of being "*understood*," which can build trust in future recommendations and increase her willingness to engage with platform suggestions (Hosey et al., 2019; Freeman et al., 2022; 2023). Conversely, if she feels her behavior is misinterpreted, she may become less inclined to offer meaningful feedback for algorithmic analysis (Cunningham et al., 2007). This interplay between Interpreters and Experiences highlights how user perception influences both system evaluation and the effectiveness of algorithmic discovery.

**Experiences ↔ Ripples**: Alex's experiences with the platform accumulate over time, producing compounding effects that extend beyond individual interactions. Her listening choices generate outcomes captured by the Ripples component. For instance, engaging with emerging artists may enhance their financial prospects and increase their exposure among other users (Bauer et al., 2017; Hesmondhalgh et al., 2023). At the same time, Alex's behavior contributes to the platform's evolving models of user patterns, reshaping

her own future experiences (Seaver, 2022). The interplay between Experiences and Ripples thus underscores how individual acts of discovery can produce broader cultural effects that, in turn, recursively shape personal musical encounters.

**Ripples ↔ Engagements**: Finally, we consider the feedback loop between Engagements and Ripples, completing the CEDC cycle. These components are differentiated primarily by their temporal position: the consequences of past discoveries (Ripples) feed into the motivations that drive future discovery acts (Engagements). As (McCourt et al., 2016) observe, music discovery is "*closely linked to configurations of musical taste that mark cultural distinctions and ongoing identity formation*." Each act of discovery refines personal taste, which in turn influences subsequent engagements (Nowak, 2016). This recursive relationship underscores how cultural contexts and individual motivations co-evolve through the dynamic interplay of discovery and identity formation.

### C.4.2. INTERNAL DISCOVERY CYCLES

In addition to the CEDC outer loop, we consider the internal connections that form between its components. These *Internal Discovery Cycles* capture direct feedback links between non-contiguous components. They emphasize that discovery is not simply a sequential progression from engagement to outcome, but a set of interlocking cycles that continuously shape one another within the discovery circuit.

**Engagements ↔ Facilitators**: Similar to the connections between Engagements and Mediations or Interpreters and Experiences, we can consider how Alex's motivation to discover music is shaped by her relationship with Facilitators. Recommender systems have become increasingly adept at modeling individual motivations, such as openness to novelty, preference for diversity, or interest in niche genres (Vargas et al., 2014; Schedl & Hauger, 2015; Mehrotra et al., 2018; Freeman et al., 2022). They are explicitly designed to align with Alex's behavioral patterns, making her motivations a core input for curatorial logic (Garcia-Gathright et al., 2018). At the same time, the design of Facilitators influences future behavior: a positive discovery experience increases the likelihood that Alex will return to the platform and may affect her broader openness to technology (Raff et al., 2020; Siles et al., 2020; Hracs & Webster, 2021). This bidirectional dynamic illustrates how user motivations not only inform curatorial practices but are also shaped by them.

**Engagements ↔ Interpreters**: Engagements also shape Alex's interactions with Interpreters. As noted earlier, Interpreters aim to capture user attributes, including inferred demographics, that influence discovery patterns (Garcia-Gathright et al., 2018; Born et al., 2021). Alex's musical preferences are codified not only through her listening behavior but also via predictive algorithms that reconstruct

demographic information from her activity (Pedersen, 2020). These interpretations, in turn, shape her future engagements. For example, socio-technical research shows that visibility metrics, such as high monthly listener counts, affect a user's likelihood of engaging with a track (Prey, 2020b; Graham & Burnett, 2019). Alex's listening behavior contributes to these metrics, which are then reflected back to her during browsing. This feedback loop illustrates how cultural valuation becomes embedded in algorithmic systems, shaping discovery through the interplay of user behavior and platform metrics.

**Mediations ↔ Interpreters**: Drawing on concepts of prominence (Mazzoli, 2020), exposure (Napoli, 2011), and prioritization (Mazzoli & Tambini, 2020), we observe how the interface placement of music is shaped by the assumptions underlying Interpreters. Since content placement is dynamic and informed by past interactions, data previously analyzed by Interpreters influences the Mediations that Alex encounters. For instance, A/B testing allows platforms to experimentally vary content placement across user groups, using Interpreter feedback to measure differential engagement and evaluate which content types are perceived as relevant (Seaver, 2022). Embedded in this process are assumptions about how positive engagement is inferred from behavioral cues (Born et al., 2021). This illustrates the bidirectional relationship between exposure and user interaction, shaped by the Mediations-Interpreters dynamic interplay.

**Mediations ↔ Experiences**: An essential dimension of musical experience lies in the medium through which listeners engage with music. The influence of Mediations on discovery Experiences is evident in user studies highlighting interface elements that users find enjoyable to use (Simon, 2020). At the same time, user experience research focuses on designing mediating technologies that anticipate and support desired user experiences (Knees et al., 2020; Petridis et al., 2022). This interplay between Mediations and Experiences forms a feedback loop connecting user agency with the design of the mediating environment.

**Facilitators ↔ Experiences**: As awareness of curatorial systems spreads among music streaming users, perceptions of algorithms increasingly take on anthropomorphic and para-social qualities (Siles et al., 2020; Freeman et al., 2023), shaping the experiences users have on a platform. In this way, Alex's understanding of Facilitators, and her broader attitude toward them, significantly shapes her experiential engagement with these systems (Morris & Powers, 2015; Hracs & Webster, 2021). Conversely, Facilitator design is informed by the types of user Experiences it aims to elicit (McKelvey & Hunt, 2019). For example, a Facilitator aiming to maximize the *hang-around factor* (Seaver, 2022) may lead to extended listening sessions within Alex's daily life. This dynamic illustrates the cyclical relationship

between Facilitator design and user experience.

**Facilitators ↔ Ripples**: The relationship between Facilitators and Ripples encompasses the impact of curatorial systems on the environments where music is consumed, commodified, and created. Facilitators are the driving force behind content curation, positioning them as *gatekeepers* between artists and users (Prey, 2020a). In response, artists often adopt strategies to counter the threat of invisibility, such as sonic optimization, i.e., deliberately crafting music to align with perceived algorithmic priorities (Morris, 2020; Morris et al., 2021). For example, many artists produce shorter songs with attention-grabbing hooks in the first thirty seconds to reduce skipping and promote sustained listening (Hesmondhalgh et al., 2023). These practices illustrate how platform-mediated discoverability reshapes musical production and artists' self-conception as cultural producers. Conversely, the conditions in which Ripples occur feed back into Facilitator design: platform value depends on a catalog that attracts users (Raff et al., 2020). If artists are unable to earn sufficient revenue, they may leave, undermining the platform's capacity to support meaningful listening experiences (Mladenov et al., 2020). This dynamic highlights the reciprocal relationship between Facilitators and Ripples.

**Interpreters ↔ Ripples**: The Interpreters-Ripples relationship reveals the entanglement of listening practices and algorithmic curation. As (Hesmondhalgh et al., 2023) note, users often adapt their behaviors to accommodate algorithms, sometimes treating these systems as sites of political contestation by interrogating their underlying logics. Such tactics include manipulating features like likes, playlist additions, or repeated plays to *train* algorithms toward preferred outputs (Freeman et al., 2022), thereby reshaping the training datasets for recommender systems (Xie et al., 2022). Conversely, the downstream effects of these discovery patterns influence Interpreter design. The well-documented feedback loop of popularity bias arises when Interpreters, trained on historical interaction data, disproportionately promote mainstream content presumed to be universally liked (Bauer et al., 2017; Henry et al., 2024; Salganik et al., 2024). This reinforcement cycle amplifies the visibility of already popular items, limiting diversity in recommendations. This dynamic underscores the need to critically examine how the consequences of past discovery patterns shape the conditions for future music discovery.

### C.5. Limitations of our Model

Our methodology is subject to several limitations. First, the search process was keyword-driven and therefore may omit relevant work that discusses discovery under different terminology. Second, while we aimed for broad coverage across technical and socio-technical traditions, the resulting corpus reflects the availability of published work and the expertise of the researchers consulted. Finally, our synthesis is qualitative and interpretive; future work could strengthen the CEDC by validating component interactions empirically and by formalizing discovery objectives and measurement strategies at multiple points of intervention.

## D. Contextualizing Current Regulatory Measures (In Detail)

Below, we provide short qualitative summaries of how each regulatory measure maps onto components of the Cultural Expressions Discovery Circuit (CEDC). We categorize components as **primary** when explicitly and directly targeted by regulatory obligations, and **secondary** when affected indirectly (e.g., as a foreseeable consequence of compliance, enforcement, or design adaptation).

We wish to note that given the breadth of regulatory measures encompassed in our work it was not possible to read every single legislative document in its original language. However, we did our best to consult verified translations and seek validation from native speakers during the mapping process.

### D.1. EU - European Union

- **Digital Services Act (DSA)**
  - **Primary components:** Mediations; Facilitators; Interpreters
  - **Secondary components:** N/A.
  - **What it covers:** The DSA regulates systemic platform risks and introduces transparency and choice obligations for recommender systems, shaping both what content is made available through curation and how it is surfaced to users. It also treats user profiling and recommender parameters as governable targets, motivating changes in user modeling and personalization pipelines.
  - **What it misses:** The DSA does not explicitly govern cultural diversity or discoverability as an outcome, nor does it directly target downstream cultural-economic impacts such as creator distribution effects. Structural barriers to discovery rooted in socioeconomic capacity or cultural participation are out of scope.
- **Digital Markets Act (DMA)**
  - **Primary components:** Mediations; Facilitators.
  - **Secondary components:** Ripples (via competition effects)
  - **What it covers:** The DMA targets gatekeeper conduct shaping visibility and access, including ranking, indexing, display, and self-preferencing practices. By constraining these prominence and ordering dynamics, it intervenes at the levels of mediations and facilitators, with indirect effects

on ecosystem structure and market access.

- **What it misses:** The DMA does not address user modeling, user experience, or cultural outcomes. Its primary objective is market contestability and fairness rather than cultural exposure diversity or discovery as a cultural process.

- **European Accessibility Act (EEA)**
  - **Primary components:** Mediations
  - **Secondary components:** Experiences
  - **What it covers:** The EEA governs accessibility requirements for digital services and interfaces, shaping how users perceive and navigate discovery environments. It primarily constrains interface-layer design choices that influence exposure and usability.
  - **What it misses:** The EEA does not directly regulate recommendation logic, user profiling, or downstream cultural-economic effects. It improves access conditions but does not prescribe discovery objectives such as diversity or novelty.

- **Audiovisual Media Services Directive (AVMSD)**
  - **Primary components:** Mediations; Facilitators
  - **Secondary components:** Ripples
  - **What it covers:** The AVMSD introduces obligations around availability and prominence of audiovisual works (e.g., European works), shaping both what content is eligible for exposure and how it is positioned within user-facing environments.
  - **What it misses:** The AVMSD does not deeply address profiling and user modeling, nor does it govern user agency and interpretability of discovery systems. Feedback loops through which compliance reshapes catalogs and engagement dynamics remain under-specified.

- **European Media Freedom Act (EMFA)**
  - **Primary components:** Mediations; Experiences
  - **Secondary components:** N/A.
  - **What it covers:** The EMFA focuses on transparency and methodological accountability around audience measurement and media ecosystems, supporting user trust and intelligibility of the media environment while indirectly shaping how visibility is structured.
  - **What it misses:** EMFA is not a recommender governance framework and does not directly regulate ranking objectives, personalization, or cultural diversity outcomes in platform discovery ecosystems.

- **General Data Protection Regulation (GDPR)**
  - **Primary components:** Interpreters
  - **Secondary components:** Facilitators; Experiences
  - **What it covers:** GDPR governs personal data processing, including profiling and inference, which directly constrains user modeling and the personalization-dependent curation systems built upon it. As such, it shapes interpreters and downstream facilitators through limits on data collection, purpose limitation, and lawful bases for processing.
  - **What it misses:** GDPR does not directly regulate interface prominence or provide explicit discoverability/diversity requirements. It reduces certain risks of personalization without ensuring improved cultural discovery outcomes.

- **EU Artificial Intelligence Act (AI Act)**
  - **Primary components:** Interpreters
  - **Secondary components:** Facilitators
  - **What it covers:** The AI Act regulates selected AI systems through risk-based obligations such as documentation, transparency, and governance controls. Where recommender or profiling systems fall within its scope, it primarily targets algorithmic decision and inference pipelines.
  - **What it misses:** The AI Act is not designed to govern cultural exposure or long-run discovery dynamics. Interface prominence, attention mechanisms, and downstream market/cultural feedback effects are largely outside its primary focus.

### D.2. UK - United Kingdom

- **Online Safety Act (OSA)**
  - **Primary components:** Mediations; Facilitators; Experiences
  - **Secondary components:** N/A.
  - **What it covers:** The OSA emphasizes safety-by-design and platform responsibility for content exposure risks. This shapes curation and ranking practices, visibility controls, and user-facing protections that affect how discovery environments are experienced and navigated.
  - **What it misses:** The OSA does not directly target cultural diversity or discoverability as cultural outcomes. It also does not specify downstream creator ecosystem objectives, leaving long-run exposure concentration dynamics largely implicit.

- **Algorithmic Transparency Recording Standard (ATRS)**
  - **Primary components:** Facilitators; Interpreters
  - **Secondary components:** Mediations
  - **What it covers:** ATRS-style standards emphasize documentation and structured transparency for algorithmic systems, including inputs, objectives, and governance processes. This makes curation systems and profiling mechanisms legible targets for oversight and auditing.
  - **What it misses:** Transparency requirements do

not directly alter prominence, attention allocation, or cultural exposure outcomes, and thus may enable compliance without changing discovery dynamics at the circuit level.

## D.3. US - United States

- **Algorithmic Accountability Act**
  - **Primary components:** Facilitators; Interpreters
  - **Secondary components:** N/A.
  - **What it covers:** Algorithmic accountability proposals typically require assessments of automated decision systems and their impacts, focusing on how models operate, what inputs they use, and how risks are managed. This maps to interpreters and facilitators as primary targets.
  - **What it misses:** These measures rarely constrain interface prominence or optimize for cultural diversity. They emphasize procedural accountability over circuit-level outcomes such as long-run exposure distribution and creator ecosystem dynamics.

## D.4. CA - Canada

- **Online Streaming Act**
  - **Primary components:** Mediations; Facilitators; Ripples
  - **Secondary components:** Experiences
  - **What it covers:** The Online Streaming Act aims to align platform ecosystems with cultural policy objectives, shaping how Canadian content is supported, surfaced, and sustained. It therefore implicates discoverability mechanisms as well as downstream cultural-economic impacts.
  - **What it misses:** It does not deeply regulate profiling pipelines or specify user modeling requirements, and its ability to translate policy goals into concrete platform-level discovery dynamics remains mediated by implementation and enforcement practices.
- **Artificial Intelligence and Data Act (AIDA)**
  - **Primary components:** Interpreters
  - **Secondary components:** Facilitators
  - **What it covers:** AIDA-style regulation focuses on responsible development and deployment of AI systems, directly targeting algorithmic decision-making and inference pipelines through governance and risk management obligations.
  - **What it misses:** It does not explicitly govern interface prominence or cultural exposure outcomes, and does not directly address market-level or cultural feedback effects on creators and catalogs.
- **PIPEDA**
  - **Primary components:** Interpreters

  - **Secondary components:** Facilitators
  - **What it covers:** PIPEDA constrains the collection and use of personal information, thereby shaping user profiling and personalization-dependent curation. It governs the informational substrate of discovery systems rather than discovery outcomes themselves.
  - **What it misses:** It does not prescribe discoverability or diversity objectives and does not regulate user interface prominence mechanisms or downstream cultural-economic impacts.
- **Broadcasting Act modernization**
  - **Primary components:** Ripples
  - **Secondary components:** N/A.
  - **What it covers:** Broadcasting modernization is primarily oriented toward cultural-economic policy (funding flows, contribution requirements, long-run ecosystem impacts), targeting ripple effects in the cultural production and distribution environment.
  - **What it misses:** It leaves platform discovery mechanisms largely unspecified, offering limited governance leverage over the interaction between interface prominence, personalization, and user modeling.

## D.5. CN - China

- **Provisions on Algorithmic Recommendations**
  - **Primary components:** Mediations; Facilitators; Interpreters, Experiences
  - **Secondary components:** N/A.
  - **What it covers:** These provisions treat recommender systems as governable infrastructure, imposing obligations that affect content ordering and exposure, user profiling practices, and user-facing controls over recommendation-driven experiences.
  - **What it misses:** The provisions do not directly govern cultural diversity or creator ecosystem distribution, and are not primarily oriented toward discovery as a cultural participation process.
- **Deep Synthesis Provisions**
  - **Primary components:** Mediations; Facilitators
  - **Secondary components:** Experiences
  - **What it covers:** Deep synthesis rules regulate synthetic/manipulated content practices and presentation, affecting both content eligibility controls and how content is framed or labeled in user-facing environments.
  - **What it misses:** The framework does not principally regulate user modeling or personalization dynamics, and it does not address cultural discoverability outcomes beyond content integrity and

governance.

## D.6. JP - Japan

- **Act on Improving Transparency and Fairness of Digital Platforms**
  - **Primary components:** Mediations; Facilitators
  - **Secondary components:** N/A.
  - **What it covers:** This law focuses on fair and transparent business practices in large digital platforms, including visibility and ranking conditions that shape access for business users. It thereby targets surface-level prominence and curation structures.
  - **What it misses:** It does not directly regulate user modeling or cultural outcomes. Feedback loops linking platform incentives to long-term creator adaptation remain outside the regulatory scope.

## D.7. IN - India

- **IT Rules (Intermediary Guidelines & Digital Media Ethics Code)**
  - **Primary components:** Mediations; Experiences
  - **Secondary components:** Facilitators
  - **What it covers:** The IT Rules impose content governance and compliance obligations that shape how content is managed and surfaced, as well as user-facing safeguards and accountability mechanisms influencing user trust and platform experience.
  - **What it misses:** The IT Rules do not robustly target personalization profiling as a governable object, and they do not provide explicit cultural diversity or discoverability objectives.
- **Digital Personal Data Protection Act (DPDP Act)**
  - **Primary components:** Interpreters
  - **Secondary components:** Facilitators
  - **What it covers:** The DPDP Act primarily governs personal data processing and consent, affecting profiling and personalization systems through constraints on data collection, processing, and retention.
  - **What it misses:** It does not regulate interface prominence or cultural exposure outcomes, and does not directly address downstream distributional impacts within creative ecosystems.

## D.8. SG - Singapore

- **Personal Data Protection Act (PDPA)**
  - **Primary components:** Interpreters
  - **Secondary components:** Facilitators.
  - **What it covers:** PDPA governs personal data practices that underpin profiling and personalization, directly shaping user modeling and recommendation-dependent discovery pipelines.
  - **What it misses:** It does not address interface prominence or define cultural discovery outcomes. Its primary objective is privacy protection rather than exposure diversity.

## D.9. KR - South Korea

- **Online Platform Fairness Act (proposed)**
  - **Date:** Proposed (2021–2024), not enacted
  - **Primary components:** Mediations; Facilitators
  - **Secondary components:** Ripples (via market competition effects)
  - **What it covers:** The proposal targets unfair ranking, self-preferencing, and discriminatory platform practices that structure visibility and access, similar to the EU DMA. It therefore intervenes primarily at the levels of mediations (how content and services are surfaced) and facilitators (ranking and eligibility mechanisms).
  - **What it misses:** It does not address user modeling, user experience, cultural diversity objectives, or circuit-level ecosystem effects beyond competition-oriented fairness framing.
- **Act on Information & Communications Network**
  - **Primary components:** Interpreters
  - **Secondary components:** Facilitators
  - **What it covers:** This act governs data handling and platform obligations that shape user profiling and automated decision systems. It constrains how user representations are produced and used for personalization.
  - **What it misses:** It does not directly regulate prominence design or provide cultural diversity/discoverability objectives, and it does not capture downstream cultural-economic feedback effects.

## D.10. BR - Brazil

- **LGPD — Brazil General Personal Data Protection Law**
  - **Primary components:** Interpreters
  - **Secondary components:** Facilitators
  - **What it covers:** The LGPD constrains profiling and datafication practices by regulating how personal data can be collected, processed, and repurposed. This directly targets user modeling and inference and, indirectly, recommender and personalization systems as they rely on regulated data processing pipelines.
  - **What it misses:** The LGPD does not explicitly govern interface prominence or discovery goals such as cultural diversity. It shapes the informa-

tional substrate of discovery but does not regulate circuit-level exposure outcomes or downstream ecosystem impacts.

- **Marco Civil da Internet (Law 12.965/2014)**
  - **Primary components:** Experiences; Engagements
  - **Secondary components:** Mediations
  - **What it covers:** The Marco Civil establishes a rights-based framework for online access and use, including principles around user rights, access conditions, and accountability norms. This primarily shapes rights framing, trust, expectations and conditions of access, and participation, with potential implications to how interface-related requirements are implemented in practice.
  - **What it misses:** It does not directly regulate curation/ranking systems or profiling mechanisms, and it does not target cultural discovery outcomes or long-term exposure distributions.

## E. Future Applications and Extensions

One promising direction concerns generative AI systems. In content discovery ecosystems, generative AI can be understood as a production mechanism that affects content supply, creator behavior, and downstream engagement dynamics. Within the CEDC framework, these effects can naturally be incorporated through components such as Facilitators and Ripples. However, platforms centered primarily on generative AI (e.g., text-to-image or text-to-music systems) differ from traditional discovery platforms in important ways. Rather than organizing access to a shared catalog of cultural content, these systems are often oriented toward on-demand generation. Consequently, the core dynamics may shift from discovery and exposure feedback loops toward generation–selection loops between prompts, generated outputs, user preferences, and platform optimization. Extending the framework to account for these dynamics represents an important direction for future work.

More broadly, similar ecosystem-level feedback structures arise in domains such as education, labor platforms, and online knowledge systems (Dean et al., 2024). We therefore see the CEDC framework as a step toward developing more general tools for analyzing how algorithmic systems shape, and are shaped by, broader socio-technical ecosystems over time.

