# OpenReview forum: "Position: Regulating Algorithms Is Not Enough. A Study of Content Discovery in Online Platforms"
_ICML.cc/2026/Position_Paper_Track — ICML 2026 Position Paper Track regular_

### Official Review · Reviewer_tyYf · 2026-03-09

**Significance:** 3
**Argument Clarity:** 2
**Rating:** 5
**Confidence:** 3

**Questions:**

* What was the criterion for inclusion of different aspects or relations in the framework? Are there alternative approaches for defining a similar framework that can make it more actionable or more concise?
* Is the framework appropriate for the analysis of platforms based on generative AI?

**Alternative Views Section:**

Yes

**Compliance With Llm Reviewing Policy A Conservative:**

Affirmed.

**Discussion Potential:**

3

**Final Justification:**

This paper provides a conceptual framework for categorizing different aspects of human-AI interaction, and applies it to study the shortcomings of current regulatory efforts. Despite its relative complexity, I feel that this framework might have potential to influence ongoing regulatory efforts, and may also have potential to influence future theoretical work at the interface of machine learning and society. The rebuttal has addressed my main concerns, and I maintain my original recommendation.

**Paper Summary:**

* This paper argues that regulating algorithms alone is not sufficient for achieving socially-beneficial outcomes in recommendation-centric online platforms, as gaps can often stem from feedback dynamics which depend on the context in which the algorithm operates.
* To address this gap, the paper proposes the Cultural Expressions Discovery Circuit (CEDC) framework, which models different aspects that should be considered when studying recommendation algorithms in a social context. The framework is analogous to the “circuit of culture” framework developed in cultural studies.
* The paper presents the different components of the framework - engagements (human inputs), mediations (UI/UX), facilitators (core algorithms), interpretation (data representation), experiences (relationship with the systems), and ripples (downstream effects). The framework is then used to contextualize current regulatory measures, and highlight their possible shortcomings. The paper concludes with a call to action for regulators and the technical research community, and a discussion of alternative views.

**Position:**

Yes

**Position In Title:**

Yes

**Related Work:**

3

**Strengths And Weaknesses:**

Strengths:
* Framework appears useful for highlighting concrete shortcomings in current regulatory efforts, and can help highlight new technical research directions.
* Points are clear, and the paper is easy to follow.

Weaknesses:
* The proposed framework appears to contain many components and possible interactions, possibly making it less effective in guiding incremental research efforts.
* The paper does not appear to discuss some economic forces that affect online platforms and may play a significant role in their development (e.g., economic incentives of platform operators, competition between different platforms, content supply and supply-side markets etc.)

**Support:**

3

---

> ### Author Rebuttal · Authors · 2026-03-30
>
> We deeply appreciate the reviewer's response to our work!
>
> ## W1: Complexity of framework
> We agree with the reviewer that our approach yields a certain level of complexity which makes acting on the entire ecosystem with one form of intervention almost impossible. However, this was never our goal. Rather, we wish to disentangle the various components so we can highlight the areas in which there should be more interventions (both regulatory and technical). As shown in our call to action (and later expanded upon in our response to ***reviewer ewpA***), our framework can be cleanly mapped to a specific domain (in our example for reviewer ewpA we propose music streaming) to yield concrete, component-level objectives for technical and regulatory bodies.
>
> ## W2: Exclusion of particular elements
> We thank the reviewer for this important observation. We agree that economic forces such as platform incentives, competition, and supply-side dynamics play a central role in shaping discovery outcomes. In the current formulation, these factors are partially captured within the Facilitators (optimization objectives and platform incentives) and Ripples (downstream effects on content supply, creator behavior, and market structure) components of the CEDC.
>
> However, we acknowledge that this connection is not made sufficiently explicit in the current draft. We will revise the paper to more clearly articulate how economic incentives and competitive dynamics shape both the design of algorithmic systems and the evolution of content ecosystems over time. Importantly, this perspective further reinforces our core argument: that discovery outcomes emerge from the complete set of ecosystem interactions, and cannot be fully understood or governed through algorithmic components alone.
>
> ## Q1: Inclusion criterion
> As described in Appendix B.2 our goal was to design a structured, unified interdisciplinary framework which is able to encompass the various conversations regarding discovery/discoverability accross many different academic domains. Thus, our inclusion criteria was related to analyzing and organizing the findings in this broad collection of literature.
>
> Finally, to our knowledge, the only other frameworks that attempts to achieve the same breadth and outcomes are [1], [2], and [3]. The first is really not geared towards the technical domain and is therefore significantly less actionable than our own. The second has, to the best of our knowledge, only been proposed, not completed. The third is oriented towards theoretical communities and is therefore neither aligned with regulatory interventions nor directly referencing creative economies.
>
> [1] McKelvey, F., & Hunt, R. (2019). Discoverability: Toward a definition of content discovery through platforms. Social Media + Society, 5(1).
>
> [2] Porcaro, L., Gómez, E., & Catarci, T. (2024). End-user algorithmic auditing for music discoverability: A research roadmap. In A. Ferraro, L. Porcaro, P. Knees, & C. Bauer (Eds.), Proceedings of the 2nd Music Recommender Systems Workshop (MRS 2024) (CEUR Workshop Proceedings, Vol. 3787). CEUR-WS.org.
>
> [3] Dean, S., Dong, E., Jagadeesan, M., & Leqi, L. (2024). Accounting for AI and users shaping one another: The role of mathematical models. arXiv. https://doi.org/10.48550/arXiv.2404.12366
>
>
> ## Q2: Applicability to GenAI
>
> We thank the reviewer for this insightful question. Our framework is primarily designed to analyze content discovery ecosystems, where users encounter and engage with a shared corpus of cultural expressions (e.g., music or video streaming platforms). In this setting, generative AI can be understood as a production mechanism within the broader ecosystem (i.e., affecting content supply and creator behavior), and can therefore be incorporated within components such as Facilitators and Ripples.
> By contrast, platforms centered exclusively on generative AI (e.g., text-to-image like MidJourney or text-to-music systems like Suno) are often oriented toward on-demand content generation rather than discovery over a shared catalog. As a result, the notion of discovery and the associated feedback loops between exposure, consumption, and cultural production may operate differently or be less central.
>
> This distinction highlights an interesting direction for future work: extending the CEDC to settings where discovery is replaced by generation–selection loops.

---

> > ### Author Rebuttal · Reviewer_tyYf · 2026-04-03
> >
> > Thanks you for the detailed response! My main concerns have been addressed.

---

### Official Review · Reviewer_ewpA · 2026-03-13

**Significance:** 3
**Argument Clarity:** 2
**Rating:** 5
**Confidence:** 3

**Questions:**

See weaknesses above.

**Alternative Views Section:**

Yes

**Compliance With Llm Reviewing Policy A Conservative:**

Affirmed.

**Discussion Potential:**

3

**Final Justification:**

I raised my score from 4 to 5 following the author's response which addressed each of my weaknesses mentioned above by providing concrete examples supporting the claims of the paper.

**Paper Summary:**

The paper argues that AI regulation should go beyond regulating the algorithms alone and move towards regulating all the individual parts of the overall sociotechnical system. The paper introduces a framework for thinking about the components of this overall system. It also argues that algorithmic regulation in tends to be limited to mitigating harms, rather than pursuing broader policy objectives of discovery and discoverability. The paper outlines different calls to actions for the technical research community and the regulatory community, and outlines different alternative view views that touch upon different aspects of the position.

**Position:**

Yes

**Position In Title:**

Yes

**Related Work:**

3

**Strengths And Weaknesses:**

Strengths:
1. The paper offers a very timely position, given the relevance and centrality of AI regulation in today's discourse, and offers very concrete calls to action for different stakeholders, including the technical research community. Many of these calls to action are substantial enough to be research agendas in themselves.
2. The paper does a good job positioning itself within an disciplinary body of work.
3. The alternative view section carefully depicts and responds to various counter arguments.
Weaknesses:
1. The arguments, while compelling, could benefit from presenting more evidence in the main body, e.g., concrete example case studies describing the limited scope of existing policies.
2. While the high-level notion of circuit design seems useful, more work could be done in justifying why the specific formulation described in CEDC can be helpful, e.g. through concrete examples that use the formulation explicitly in the call to action section.
3. The alternative view section reads more like soft counter arguments rather than flushed out alternative views. While the counter arguments are strong, it could help to have them explicitly propose an alternative (e.g., status quo) rather than just question the paper's proposal.

**Support:**

3

---

> ### Author Rebuttal · Authors · 2026-03-30
>
> ## W1: Inclusion of further case studies
> While we do not include these examples in our main body, we provide a detailed analysis of each regulatory body from Table 1 in Appendix C. For example, at the EU level, current regulatory frameworks primarily target algorithmic curation and interface design. These gaps are not only structural but observable in practice. Transparency-focused regulations such as the Digital Services Act (DSA) require platforms to disclose information about recommender systems and ranking logic. While this improves visibility, it does not directly alter content presentation or address upstream factors (e.g., user engagement patterns) and downstream effects (e.g., creator incentives). As a result, even under formal compliance, exposure often remains highly concentrated, with already-popular content dominating attention and limiting discovery of independent or local artists.
>
> If the reviewer could provide further suggestions on the case studies to be studied, we would happily augment the information to improve our paper.
>
> ## W2: Tying components to the call to action
> We thank the reviewer for this helpful suggestion and will include a case study in the revised version. While the current structure aims for generality across modalities, a domain-specific example clearly illustrates how the framework can be operationalized. For example, in a music streaming context, the CEDC can be used to identify complementary interventions across components, each with aligned technical and regulatory objectives:
>
> ### Engagements:
> * Technical goal: Model and account for heterogeneity in user readiness for discovery (e.g., differences in cultural participation as a function of infrastructure and resources).
> * Regulatory goal: Support cultural participation through investments in local music ecosystems (e.g., community programs, education, or funding for local scenes).
> ### Mediations:
> * Technical goal: Optimize interface design for balanced exposure (e.g., prominence-aware ranking and layout evaluation).
> * Regulatory goal: Ensure fair and non-manipulative presentation of content (e.g., constraints on prominence, avoidance of dark patterns).
> ### Facilitators:
> * Technical goal: Develop recommendation algorithms that balance relevance with diversity and long-term exposure.
> * Regulatory goal: Impose accountability on ranking systems (e.g., transparency, auditing, or diversity-related obligations).
> ### Interpreters:
> * Technical goal: Learn richer user representations that capture exploratory intent and dynamic preferences.
> * Regulatory goal: Govern user modeling practices (e.g., limits on profiling, requirements for user control and interpretability).
> ### Experiences:
> * Technical goal: Design systems that support user agency and exploration (e.g., controllable or interactive recommendation modes).
> * Regulatory goal: Ensure user rights to transparency and control over recommendation systems (e.g., ability to adjust or opt out of personalization).
> ### Ripples:
> * Technical goal: Model and evaluate long-term ecosystem effects (e.g., impact on content supply, creator incentives, and distributional outcomes).
> * Regulatory goal: Address downstream cultural and economic effects (e.g., requirements for data access, funding mechanisms, or monitoring of exposure outcomes).
>
> Together, these interventions illustrate how discovery outcomes emerge from coordinated changes across components, rather than from algorithmic adjustments alone.
>
> ## W3: alternative views
> We thank the reviewer for this suggestion and will revise Section 7 to more explicitly present alternative paradigms. In particular, we will frame algorithm-centric governance as the dominant status quo, including its motivations and strengths, before discussing its limitations.
>
> Concretely, algorithm-centric governance treats ranking and recommendation systems as the primary levers for improving platform outcomes. This perspective is attractive because it is technically tractable, measurable, and aligns with existing regulatory tools such as transparency, auditing, and fairness constraints. Under this view, modifying ranking objectives (e.g., to promote diversity or mitigate popularity bias) is expected to improve discovery through feedback loops between users, creators, and the platform.
> However, this approach assumes that system-level outcomes can be shaped through algorithmic adjustments alone. In practice, such interventions operate on observed interaction data and act only indirectly on upstream and downstream factors (e.g., user engagement conditions, creator resources, and production incentives). As a result, they may be insufficient to achieve sustained improvements without complementary interventions across the broader ecosystem.

---

> > ### Author Rebuttal · Reviewer_ewpA · 2026-04-03
> >
> > Thank you for the thorough response. The concrete examples are very helpful. Adding all of these to the main paper with greatly strengthen the paper. I have updated my score accordingly.

---

### Official Review · Reviewer_QYME · 2026-03-16

**Significance:** 3
**Argument Clarity:** 3
**Rating:** 5
**Confidence:** 4

**Questions:**

1. The call for quantification makes sense being a computer scientist. But often this is our limited world view. For example, how to model the emergent cultural behaviours? People adapt to AIs and other digital platforms. How can we realistically model them or can we at all? How do the paper envision the usefulness of CEDC in that case?
2. I do not comprehend the point 2 of alternative views-- specially, the statement "However, the CEDC is intended to prevent such interventions from being treated as a complete solution rather than a partial one." Why does CEDC have to be contradictory to algorithmic interventions? Why cannot it incorporate algorithmic interventions as a subset? Can you please explain.
3. In the title, the cultural perspective is not highlighted. It looks a bit more generic than the content. If possible, it would be nice to clarify this point.
4. In context of education, I found similar arguments and perspective to be discussed in some chapters (Part D-F) of the book "AI Bias in Education: Performing Critical Oversight: Perspectives and Practical Approaches for Educators". Though the authors focus on online platforms, a brief discussion on its connection to other applications like education would be interesting for the readers.

**Alternative Views Section:**

Yes

**Compliance With Llm Reviewing Policy A Conservative:**

Affirmed.

**Discussion Potential:**

4

**Final Justification:**

With the changes of adding the concrete example and extending the discussions mentioned in rebuttal, I would like to retain my positive perspective.

**Paper Summary:**

This paper points out the weaknesses of the present algorithm-focused regulations. In this context, it yields a generic definition of discovery and further introduces the conceptual and interdisciplinary framework of Cultural Expressions Discovery Circuit (CEDC). To elaborate CEDC, the paper identifies six components, and discusses interactions between them and the corresponding emergent behaviours. Finally, it voices a call for action to enable CEDC and also briefly discusses its plausible limitations in the alternative views section.

**Position:**

Yes

**Position In Title:**

Yes

**Related Work:**

3

**Strengths And Weaknesses:**

Strengths:
1. The paper is well-written with well-thought components and concepts.
2. The discussion on limitation of present algorithm-focused regulations are useful and timely.
3. The importance of incorporating culture, and the components, interactivity, and emergent behaviours identified for developing CEDC are at point and well argued.

Weaknesses:
The alternative section is brief. I have some further questions about them.

**Support:**

3

---

> ### Author Rebuttal · Authors · 2026-03-30
>
> We are deeply thankful for the reviewer's positive response to our work!
>
> ## Q1: Quantification of ambiguous cultural phenomenon
> We agree with the reviewer that emergent cultural behaviors cannot be fully captured by formal models. The CEDC is therefore designed to support an explicitly interdisciplinary approach: combining granular socio-technical methods (e.g., HCI, anthropology, and sociology, which are particularly well-suited to capturing edge cases, situated practices, and non-formalizable behaviors) with aggregate empirical analyses (e.g., large-scale interaction data and economic indicators).
>
> Its usefulness lies in providing a unified vocabulary that situates these different forms of evidence within a shared system, making explicit how insights from one component (e.g., user behavior or creator incentives) propagate through the ecosystem and affect algorithmic outcomes. In this sense, the CEDC does not require fully specified models of cultural behavior, but instead enables structured interdisciplinary reasoning in settings where such behaviors are inherently difficult to formalize. The empirical metrics highlighted in our call to action reflect the types of contributions that the technical community is well-positioned to make, rather than an assumption that these metrics alone are sufficient to capture the full complexity of the system.
>
> ## Q2: Contradictions with algorithmic interventions
> We appreciate the opportunity to clarify what we mean by raising this alternative view. As exemplified by the writing of ***reviewer bmhf*** there are those who may argue that algorithmic interventions are sufficient for addressing many of the regulatory issues associated with discovery of creative content. Our goal is to emphasize that algorithmic elements address only part of the system. By making other components (e.g., engagement and ripple effects) explicit, the framework helps identify complementary interventions that operate on the data-generating process. We will happily revise the text to clarify that the CEDC subsumes and complements algorithmic approaches, rather than positioning them in opposition.
>
> ## Q3: Title
> We thank the reviewer for this suggestion and, if the camera ready policy allows it, we will happily revise the title to better reflect the cultural and socio-technical focus of the paper (e.g., emphasizing discovery of cultural expressions).
>
> ## Q4: Broader implications of our framework.
> We appreciate this suggestion and agree that the perspective extends beyond online platforms. The CEDC framework is intended to capture general properties of sociotechnical systems in which algorithmic components interact with human behavior and institutional context. For example, in educational settings, similar dynamics arise through feedback between learning platforms, student behavior, and institutional practices. We will happily add a brief discussion to highlight these connections and clarify the broader applicability of the framework.

---

> > ### Author Rebuttal · Reviewer_QYME · 2026-04-07
> >
> > Thanks for the response. With the changes of adding the concrete example and extending the discussions mentioned in rebuttal, I would like to retain my positive perspective.

---

### Official Review · Reviewer_bmhf · 2026-03-16

**Significance:** 2
**Argument Clarity:** 2
**Rating:** 3
**Confidence:** 3

**Questions:**

Given algorithms and data are key important pieces in online platforms, what objectives exactly cannot be achieved by regulating algorithms and data? For example, culture and content diversity can be achieved by various exploration algorithms in recommendation systems considering the feedback loop among system, content creators and users.

**Alternative Views Section:**

Yes

**Compliance With Llm Reviewing Policy A Conservative:**

Affirmed.

**Discussion Potential:**

2

**Paper Summary:**

This paper advocates to look beyond regulating algorithms for content discovery in online platforms. The authors argue that objectives such as content discovery diversity cannot be achieved only with algorithm-centric approaches alone. Instead they propose to look at the interactions among models, interfaces, user behaviors, economic incentives and culture norms besides algorithms under the Cultural Expressions Discovery Circuit (CEDC) framework. Under this framework, the authors discuss how certain regulatory approaches can struggle to align with broader objectives.

**Position:**

Yes

**Position In Title:**

Yes

**Related Work:**

2

**Strengths And Weaknesses:**

Strengths:
This paper studies important aspects of content discovery in online platforms.

This paper outlines key elements of content discovery in online platforms, with the proposed CEDC framework.

This paper makes some valid point on efforts strengthening engagements and other elements of the entire ecosystem.

Weaknesses:
This paper lacks of clarity and relatively hard to follow. Despite of making valid points, this paper didn't clearly discussing what else can be done and significantly missing beside regulating algorithms. It became somewhat reasonable after Table 1 outlined what has been missing in existing regulation. But these points are not mentioned clearly in the first half of the paper. For example, in the music streaming example provided in Introduction, it is unclear what is beyond algorithm-centered analysis and why this unclear method can enable identifying how platform design and exposure patterns induce strategic adaptation by creators.

Even though the entire eco-system has much more than algorithm and data, online platform are user-centric enabled by algorithms and user generated data. Therefore the design of the algorithm and their regulations can easily affect the entire ecosystem. For example, the modification of algorithms provide incentives of infrastructure for content creators and local organizations. It is unclear how significant new policies and support on other aspects compared to regulating the algorithms.

**Support:**

2

---

> ### Author Rebuttal · Authors · 2026-03-30
>
> ## W1: Clarity and positioning of extra-algorithmic interventions
>
> We thank the reviewer for the thoughtful feedback. Our intent was to first introduce the CEDC components (Section 3) as a shared vocabulary for reasoning about ecosystem dynamics, before discussing regulatory gaps (Section 4).
>
> We will happily update the introduction to explicitly explain the importance of extensions beyond algorithm-centered analysis. To achieve this, we can introduce examples of non-algorithmic factors captured by the CEDC framework, highlighting the critical role that infrastructure and resource availability for artists play in the discovery process. We will also revise the music streaming example to more clearly illustrate the importance of socio-technical analysis. For more details on these revisions, please see our response to Q1 below.
>
> We are confident that these updates will make the role of extra-algorithmic (socio-technical) factors clearer earlier on in the paper.
>
>
> ## W2: Why algorithmic interventions are insufficient
> We agree that algorithmic design is a powerful lever in online platforms and can influence incentives within the ecosystem. However, our claim is that algorithms operate on observed data but do not directly control how this data is generated.
>
> Even with advanced algorithmic approaches (e.g., re-ranking or exploration strategies), such interventions primarily affect the allocation of exposure over existing content. For example, re-ranking content to promote independent artists can increase short-term exposure, but does not address structural constraints such as limited production resources, lack of audience information, or coordination barriers among creators [1, 2, 3]. These factors directly influence what content is created and how users engage with it, thus shaping the data distribution on which algorithms operate.
>
> As a result, while algorithmic interventions can influence behavior within a given content supply, they cannot by themselves ensure sustained changes in content production or representation. Objectives such as sustained representation of new or independent artists therefore require complementary interventions that act on the data-generating process itself.
>
> Our position is not to downplay the importance of algorithms, but to highlight that achieving these objectives requires jointly considering algorithmic and extra-algorithmic (socio-technical) levers.
>
>
> ## Q1: Concrete example (music streaming)
> In music streaming, the distinction between algorithmic and socio-technical interventions becomes clearer. Algorithmic approaches to discovery (e.g., mitigating popularity bias through re-ranking or exploration) operate on the set of content already available on the platform and can improve exposure diversity within this existing catalog.
>
> However, music discovery objectives, such as sustained diversity of content production or long-term representation of independent or local artists, depend on how content is created and how creators and users interact with the platform over time. These aspects are shaped by factors such as production resources, access to audience information, and coordination within artistic communities, which are not directly controlled by algorithms.
>
> In contrast, engagement-level interventions (e.g., supporting local artistic ecosystems or community outreach) and ripple-level interventions (e.g., providing artists with actionable audience analytics) directly influence both content supply and user behavior. Notably, platforms themselves employ such extra-algorithmic mechanisms (e.g., subsidizing production resources for independent artists) [4], highlighting that discovery outcomes are shaped by broader ecosystem dynamics rather than algorithmic design alone.
>
>
> [1] G. Born, J. Morris, F. Diaz, A. Anderson, Artificial intelligence, music recommendation, and the curation of culture, Technical Report, The Schwartz Reisman Institute for Technology and Society (SRI), 2021.
>
> [2] Hesmondhalgh, D., Campos Valverde, R., Kaye, D. B. V.,
> and Li, Z. The Impact of Algorithmically Driven Recommendation Systems on Music Consumption and Production: A Literature Review.
>
> [3] Prey, R. Performing Numbers. The Performance Complex: Competition and Competitions in Social Life, pp. 241, 2020. Publisher: Oxford University Press Oxford.
>
> [4] https://artists.spotify.com/spotify-music-studios

---

> > ### Author Rebuttal · Reviewer_bmhf · 2026-04-05
> >
> > Thank you for the response! Some of my concerns has been addressed. I agree that regulating algorithms is not enough hence this paper studies an important matter. I still feel the organization and proposed approach lack of strong support, hence keeping my score.

---

### Decision · Program_Chairs · 2026-04-30

**Decision:**

Accept (regular)

**Comment:**

The paper presents an important and timely position. It clearly supports that position through discussions and examples, and presents and discusses the alternative view too. Reviewers found that the topic of content discovery and regulation is important, the paper is well-written and presents well-thought concepts.
The concerns that the reviewers brought up relate to some required clarifications, better organization, better justification of the specific CEDC framework, required strengthening of the alternative views section, and more concrete examples to strengthen the evidence behind certain claims. During the rebuttal, the authors were able to address most concerns, with most reviewers recommending acceptance. In particular, their rebuttal illustrated how "extra-algorithmic" factors such as artist resource availability and community infrastructure are critical to discovery outcomes that re-ranking algorithms alone cannot sustain. In the rebuttal, the authors also walked through a music streaming example, to illustrate these issues more concretely. The expanded alternative views section makes a stronger case behind the status quo perspective before showing the contrast with the position argued for in this paper. The authors also discussed issues with quantification of cultural phenomena that they identify, and how the proposed CEDC can support interdisciplinary discussions despite these issues, by offering unifying vocabulary. Finally, to address issues of complexity of the proposed framework and lack of justification of its components, the authors discuss how to map the framework to a specific domain (music streaming example) to illustrate its usefulness.
Overall, the consensus among reviewers is positive, with the exception of organization issues of the paper which I feel can be addressed and shouldn’t hold it back from publication. The paper addresses a timely and important issue that is likely to stimulate discussion at the conference.